# Assessing coupling interactions in a safe and just operating space for regional sustainability

Dongni Han[1], Deyong Yu [1,2,3] ✉ & Jiangxiao Qiu [4]

Human activities affect the Earth System with an unprecedented magnitude, causing undesirable irreversible degradation. The United Nation's Sustainable Development Goals (SDGs) provide an integrated global action plan for sustainable development. However, it remains a great challenge to develop actionable strategies to achieve regional sustainability within social-environmental constraints. Here we proposed a framework, integrating safe and just operating space (SJOS) with SDGs, to assess regional sustainability and interactions between environmental performance and human well-being across scales. Despite China has not fully achieved sustainable development from 2000 to 2018, most provinces have shown significant improvements. Our analyses further delineated four development patterns (i.e., coupled and developed, coupled and underdeveloped, uncoupled and underdeveloped, and coupled and underdeveloped), and developed targeted strategies and pathways for each pattern to transition towards sustainability. Our operationalizable framework is broadly applicable to other regions or nations to actualize sustainable development.

Since the industrial revolution, the Earth System has entered the Anthropocene, where human activities have been the predominant driver of global environmental changes[1]. In consequence, there are looming signs that several critical thresholds in resource use, emissions and environmental degradation are being approached or even transgressed (e.g., biosphere integrity, climate change, land-system change and biogeochemical flows)[2–4]. Substantial evidence has indeed demonstrated that the Earth System is moving towards an unsustainable trajectory or into a state undesirable for humanity to safely operate[5]. The search for transition towards sustainability is therefore urgently needed yet remains a pivotal challenge in the Anthropocene.

The Sustainable Development Goals (SDGs) provide a more integrated and inclusive solution to sustainable development, presenting a paradigm shift from a conceptual agenda to measurable standards and operationalizable transformations[6,7]. Nevertheless, the assessment of

sustainability and policy tracking and monitoring of progress towards SDGs remain challenging with often fragmented, isolated and inconsistent implementation. In some worse situations, even if SDGs are achieved, the environment may be further degraded[8]. Thus, there is a pressing need to understand how to quantify and assess the progress towards achieving SDGs to inform actionable policies and sustainable development strategies.

This underlines the scientific and practical need of developing robust, accurate, and comprehensive assessment framework to guide sustainable development[9–11]. The framework should identify the sustainability gaps between target social-ecological system's actual performance and corresponding sustainability standards[12]. The standards can be either set forth by policy targets or derived from identified capacity thresholds, including both environmental limits and social thresholds[13]. To that end, a set of sustainability standard concepts have

[1]State Key Laboratory of Earth Surface Processes and Resource Ecology, Faculty of Geographical Science, Beijing Normal University, Beijing 100875, China. [2]Key Laboratory of Tibetan Plateau Land Surface Processes and Ecological Conservation, Qinghai Normal University, Xining 810016, China. [3]Academy of Plateau Science and Sustainability, People's Government of Qinghai Province and Beijing Normal University, Xining 810016, China. [4]School of Forest, Fisheries, and Geomatics Sciences, Fort Lauderdale Research and Education Center, University of Florida, Davie, FL, USA. ✉e-mail: ydy@bnu.edu.cn

been developed, such as limits-to-growth[14], safe minimum standards[15,16], precautionary principle[17], and tolerable windows[18]. Planetary boundary (PB) framework[19,20] is an emerging concept that builds on and enriches the previous sustainability standards. A key advance is that the PB framework focuses on the biophysical processes of the Earth System that determine the self-regulating capacity of the planet[19]. PB framework[19,20] proposes quantitative limits for the anthropogenic appropriation of the Earth's providing capacity and delineates a safe operating space for humanity. Beyond such PB limits, abrupt or irreversible environmental changes would be deleterious or even catastrophic for human society. To account for socio-economic dimensions, the safe and just operating space (SJOS) framework[21] has been developed that further integrates PB (i.e., biophysical limits)[19] with the social foundations (i.e., basic human needs)[22] to assess the sustainability standards of social-ecological systems[23].

Despite recent advances of SJOS framework in assessing sustainability, several prominent gaps exist. (1) Transferrable downscaling. Sustainable development is generally implemented by governments, corporates, communities, and keystone actors operating at respective national, sub-national, regional, and local scales[24]. Policy-oriented assessments thus need to downscale the framework so that it is capable of addressing sustainability issues spanning across scales (from local to regional, national, and planetary) and socio-ecological contexts[25]. Much prior work has attempted to translate the SJOS framework to national or regional scales using different approaches[26–28]. Yet joint implementation of environmental footprints and PB of SJOS in the footprint-boundary framework[9] to assess environmental sustainability remains scarce. Such an approach, nonetheless, provides a scalable, replicable, and transferrable path to quantify environmental sustainability of SJOS that can serve as a benchmark to assess progress[29–31]. (2) Spatial-temporal dynamics. In assessing sustainability relative to SJOS, most prior studies are either static (assessment for a given time period) or focusing on temporal changes in region-wide summaries. Few research has explicitly addressed spatial heterogeneity and temporal dynamics in environmental performance and human well-being[32]. However, such information can be crucial for identifying hotspots for targeted policy actions, for understanding drivers leading to spatial variations in sustainable development, and for tracking progress towards achieving SDGs. (3) SDG interactions. It is increasingly revealed that there are complex interactions (i.e., trade-offs, synergies) among different SDGs[33,34]. Hence, it is critical to consider both biophysical processes and social well-being as well as their interactions in assessing sustainability based on SJOS, which thus far remains less well addressed.

Hence, in order to account for spatial-temporal dynamics and intricate interactions between environmental performance and human well-being goals in sustainability assessment, we introduced an innovative 'coupling coordination degree' (CCD) to the SJOS analytical framework, a concept originally derived from Physics[35]. "Coupling" refers to the phenomenon that two or more systems interact with each other closely in various ways. "Coordination" reflects the degree of coherence between subsystems, as well as the extent to which the system tends to move towards the desired order. Hence, CCD is a measure of the synergies among interacting subsystems, which determines the trajectory of integrated social-ecological system from disorder to order[36]. For example, a CCD of 1 reflects the development of perfect coherence, whereby all subsystems are synergistic. This concept has been widely adopted to measure the interactions between two or more (sub)systems, especially in relation to conflicts between environment and economic dimensions, such as between urbanization and eco-environment[37], economic development and ecological environment[36], ecosystem services and urban development[38,39]. It can also be used to evaluate and compare the emergent, or system-level outcomes from policy interventions (such as public investments and regulations).

In this research, to demonstrate the application of CCD in the SJOS framework for sustainability assessment, we focused the actionable case study for China—the world's most populous country and largest exporter, as well as the key actor in global sustainability. China provides resource base and produces goods for other countries[40], and bears the environmental impacts induced by consumption in other parts of the world[29]. Besides, many previous socio-economic developments have come at the expense of degrading environment, posing serious conflicts between economic development and ecosystem stewardship. Human exploitation of resources, socio-economic reforms, rapid population growth, industrialization and urbanization have also accelerated this long-term conflict[41]. Therefore, China is facing significant and urgent challenges in ensuring sufficient available resources that are used to meet the needs of all. That is, the nation should emphasize the sustainable use of regional resources for human well-being. China also embraces substantial spatial heterogeneity in natural resource, cultural heritage, environmental integrity and social inequality, and undergoes >30 years (since 1986) of long-term sustainable development practices, presenting an ideal case for testing our proposed sustainable assessment framework.

The three main objectives of this work are as follows: (1) to assess sustainable development status in the context of SDGs by measuring the performance relative to a defined SJOS; (2) to implement the CCD in SJOS for identifying development patterns using sustainability assessment (i.e., from objective 1) that considers human-environment interactions at multiple scales in a spatially explicit manner; and (3) to put forward targeted strategies on development patterns as their progress towards regional sustainability.

In social-ecological systems, interactions between human and biophysical systems are bidirectional and determine the dynamics of the overall system. On one hand, a healthy ecosystem is fundamental to support the sustainable development of humanity, which provides basic materials and services for human survival and economic development[42]. On the other hand, human development can provide capital guarantee, infrastructural and technological support to conserve the environment. However, intensive resource consumption, land-use change, and pollutions may have negative effects on the environment[43].

Hence, the conceptual foundation of this research integrates the SJOS and SDGs frameworks to investigate regional sustainability in social-ecological systems that explicitly considers the interactions between human and biophysical systems using CCD model (Fig. 1). To achieve sustainable development, human society must operate within the SJOS—the sustainability range between environmental limits set forth by PBs and environmental footprints, and social foundations defined as minimum standards or targeted social thresholds of human outcomes, while environmental performance and human well-being are acting in a synergistic development pattern.

Specifically, our conceptual framework analyzed coupling relationships between environmental performance and human well-being and development patterns of social-ecological systems in ways that can be explicitly applied to inform sustainable management. Based on our conceptualization, development patterns are divided based on two dimensions: the level of coupling and the level of development (Fig. 2). The coupling level is quantified by the magnitude of CCD, whereas the development level is quantified by changing trends of CCD. Across the coupling level (y-axis), the regions or systems can be categorized as coupled or uncoupled. High levels of coupling indicate synergies (coupling), whereas low values indicate trade-offs (decoupling) between achieving environmental performance and human well-being goals. Along the development level (x-axis), regions or systems can be classified as developed or underdeveloped. Developed regions tend to have an increasing level of coupling (more coupled), whereas underdeveloped regions show a trend towards a more uncoupled direction (more uncoupled). Hence, these two dimensions delineate

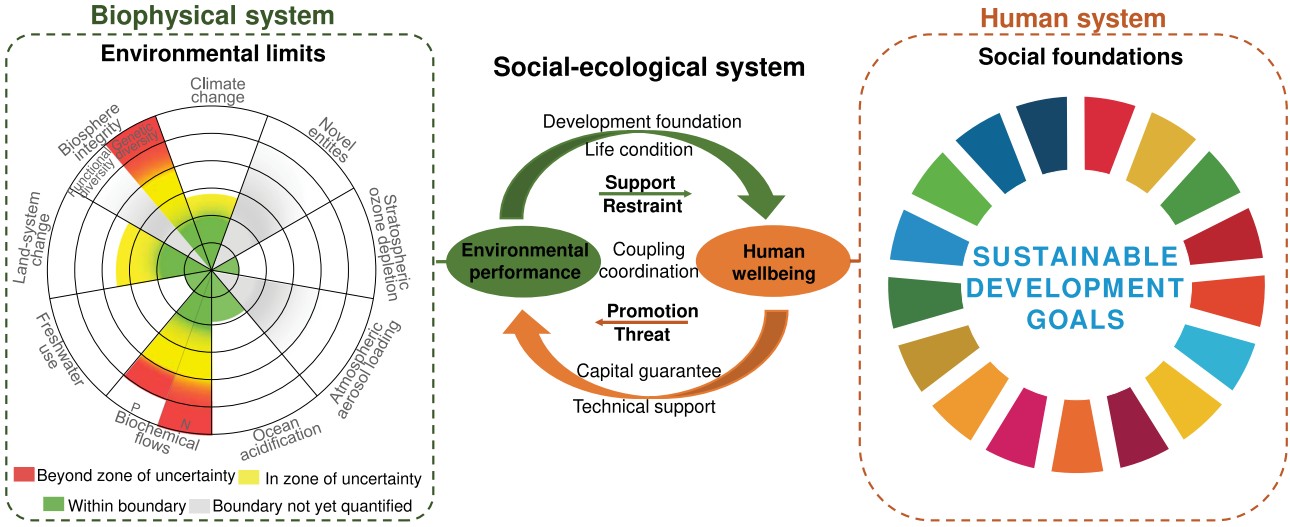

**Fig. 1 | Conceptual framework of the coupling coordination relationships between environmental performance and human well-being.** The framework integrates the safe and just operating space and footprint-boundary frameworks to measure the sustainability related to the corresponding SDGs within a coupled social-ecological system. Human well-being indicators (social foundations within human system) are corresponding to the SDGs, while the processes to achieve these goals should be restricted by the environmental limits (biophysical system), namely environmental limits quantified by the downscaled planetary boundaries. These two subsystems are not isolated, but there are complex interactions between them.

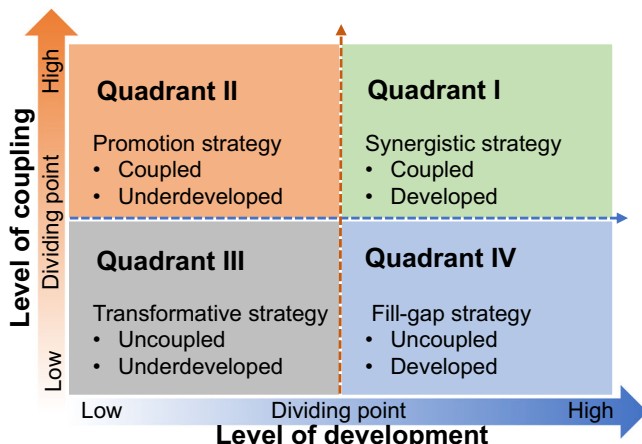

**Fig. 2 | Development patterns and corresponding sustainable management strategies towards a more sustainable development direction.** Four development patterns are divided based on two dimensions: level of development (*x*-axis) and level of coupling (*y*-axis). The level of coupling classifies regions according to the magnitude of coupling coordination degree between human well-being and environmental performance, whereas the level of development reflects changing trends in coupling coordination degree over time. Regions located in the quadrants away from the middle horizontal line are more coupled. Regions in the quadrants far away from the middle vertical line are more developed.

four quadrants of development patterns that can be used for generating corresponding sustainable management strategies. Quadrant I, as the relatively ideal development pattern, performs well both in coupling and development dimensions. Such development pattern in Quadrant I indicates that the prerequisite for sustainable development requires synergies between environmental and socio-economic aspects. Quadrant II is for coupled and underdeveloped pattern, with the coupling level above the dividing point and the development level below the dividing point. Whereas Quadrant III represents uncoupled and underdeveloped pattern, whose coupling and development levels are both below the dividing points. Wherein coupling levels of given regions are classified into Quadrant III, it indicates that management

policies cannot meet the requirements of sustainable development[44]. In Quadrant IV, the regions within are uncoupled but well developed.

## Results

To implement our conceptual framework, we defined the SJOS that lies between environmental limits and social foundations in the context of SDGs. Investigating the status of a focal regional or system relative to the defined SJOS helps us: (1) to confirm whether it is possible to operate below the Earth's carrying capacity without compromising essential social welfare; (2) to quantify the coupling coordination degree between environmental and socio-economic performance (i.e., level of coupling) and its changes (i.e., level of development); and (3) to develop possible policies and strategies and set up realistic expectations on how to best satisfy human basic needs in a sustainable manner.

This section presents the results of these three interrelated analyses mentioned above (i.e., sustainability assessment, coupling interaction, and policy and strategy development), with their implications and associated conclusions discussed in the subsequent sections.

### Assessment of sustainability performance based on SJOS

We gathered historical data from 2000 to 2018 and analyzed national performance on five environmental footprints (relative to downscaled PBs) and 10 social indicators (relative to social foundations), informed by the SJOS (Fig. 3).

For environmental performance, at both the global and national scale, three of the five environmental boundaries have been substantially overshot (climate change, phosphorus, and nitrogen cycles), considered as high risk status. Whereas the other two processses (land-system change and freshwater use) still stay within the boundaries, regarded as safe status. Generally, China's performance on environmental dimension is worse than the global level, except for land-system change. In particular, climate change, phosphorus and nitrogen cycles have exceeded their boundaries by 3.86, 9.92, and 3.68 times in China (Table S11).

For social performance, from a global perspective (Fig. 3a), 1 out of the 10 social indicators has reached the thresholds (i.e., jobs). In contrast, for China, at the national scale (Fig. 3b), the thresholds have been achieved for 2 indicators (i.e., energy and jobs), and the country also performs well in food security, income, and education. In

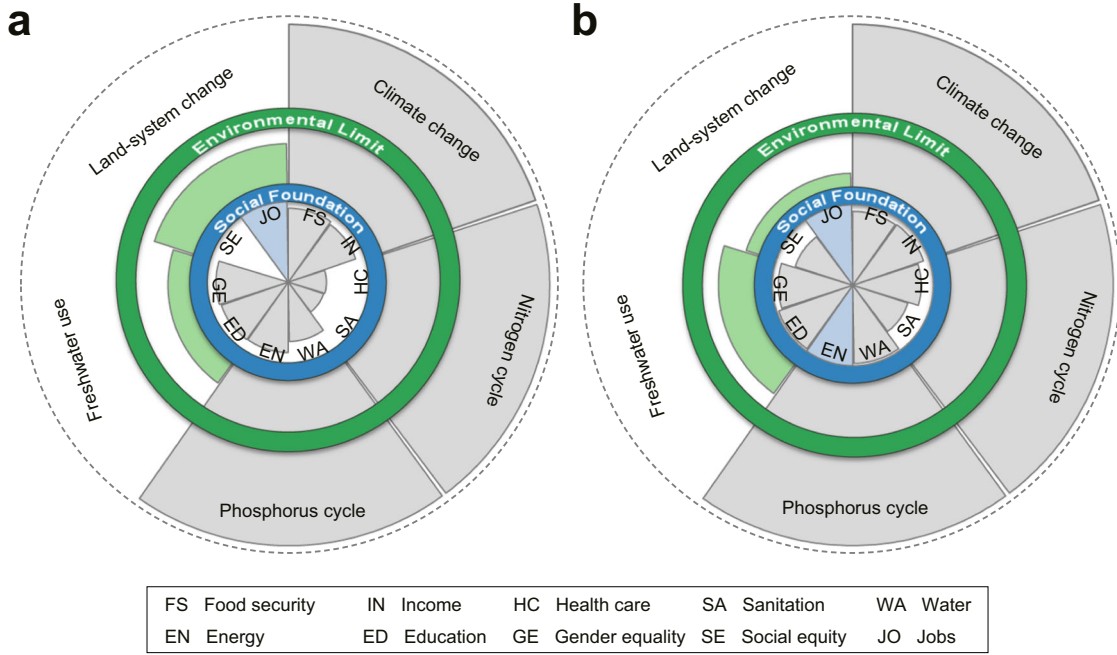

| FS | Food security | IN | Income | HC | Health care | SA | Sanitation | WA | Water |
| EN | Energy | ED | Education | GE | Gender equality | SE | Social equity | JO | Jobs |

**Fig. 3 | The performance relative to the safe and just operating space from 2000 to 2018. a** The world. **b** China. When internal wedge reaches the social foundation and external wedge is within the environmental limit, the state is considered a safe and just operating space. Internal wedges indicate actual social indicators relative to the social foundations. External wedges show environmental footprints relative to the environmental limits. Values depict the average of the specific indicators. The wedges measure the status for each dimension as a percentage compared to its boundary (0% at the center and 100% at the boundary). The environmental limits respected are green wedges, and social foundations reached are blue wedges. Wedges with a dashed extend beyond the chart area. Adapted from O'Neill et al.[30]. See Tables S6 and S7 for the specific data sources.

general, compared to the world, China has a higher level of social outcomes relative to social foundations, except for gender equality (Table S12).

In tandem, China did not entirely operate within a SJOS. Compared to the global average, China underperforms in environmental performance, yet outperforms in human well-being. This may be attributed to the fact that China provides the resource base for other countries in international trade and thus bears the environmental consequences of outsourcing. In this process, China has traded off economic development and improvements in human well-being, at the expenses of environmental impacts[45], such as $CO_2$ emissions and land-use conversions.

In addition, to get a clear picture of China's social progress and ecological degradation over time and space (Figs. 4 and 5), we traced the spatial and temporal trajectories of environmental and social performance in the period 2000–2018. To measure the environmental performance, four PBs (climate change, land-system change, freshwater use, and biogeochemical flows) are downscaled to per capita shares and compared to corresponding environmental footprints (Fig. 4a). As two indicators are measured for the biogeochemical flow of PB (i.e., nitrogen and phosphorus cycles), five environmental indicators are thus considered. Environmental performance indicates the ratio of environmental footprints to environmental limits (i.e., downscaled PBs). The footprint-to-boundary ratios depict whether the Earth's carrying capacity has already been exceeded. The control variables we selected are annual $CO_2$ emissions, surface of anthropized land, freshwater use, and allocation of nitrogen and phosphorus fertilizer applied to cropland, respectively. Per capita environmental limits from 2000 to 2018 are listed in Table S10.

Substantial spatial heterogeneity occurs in environmental performance across provinces. At the provincial scale (Fig. 4a), most China's provinces significantly exceed the per capita values, using resources at levels above the environmental limits. Among them, the most challenging boundary to confer is climate change: no provinces are within the boundary. In addition, the boundaries of nitrogen and phosphorus cycle have also been exceeded for most provinces. The percentages of provinces that are within the per capita boundaries of land-system change, nitrogen and phosphorus cycles are 36.7%, 6.7%, and 6.7%, respectively. While the situation for freshwater use is considerably better, with 80% of the provinces staying within the limits. We found that none of the provinces operate within all biophysical boundaries at the same time. Our analysis revealed significant disparities in numbers of boundaries respected, ranging from zero to four.

After analyzing the spatial distribution of environmental performance, we next examined the temporal evolution from 2000 to 2018 (Fig. 5a). At the national scale, all environmental footprints of China have increased in the period 2000−2018, moving away from the SJOS (Table S13). Overall, there has been an increasing trend in environmental pressure across provinces over time. Specifically, $CO_2$ emissions of all provinces have significantly increased, except for Beijing. Blue water footprint, land footprint, nitrogen footprint and phosphorus footprint have increased over time in 70%, 73%, 70% and 73% of the provinces, respectively. Notably, decrease in environmental footprints is predominantly located in the eastern provinces of China, such as Beijing-Tianjin-Hebei and eastern coastal areas.

Our results showed that environmental performance relative to environmental limits (i.e., downscaled PBs) varied greatly across indicators. The differences depended on the types of environmental indicators, such as resource-based indicators (e.g., freshwater use, land-system change), versus pollution-based indicators (e.g., climate change, biogeochemical flows). This suggests that appropriate policies need to be developed for each PB, considering its importance and characteristics.

To quantify human well-being as proxies for social foundations, we selected 10 social aspects following the SDGs and SJOS frameworks. For each aspect, we chose the corresponding social indicator and

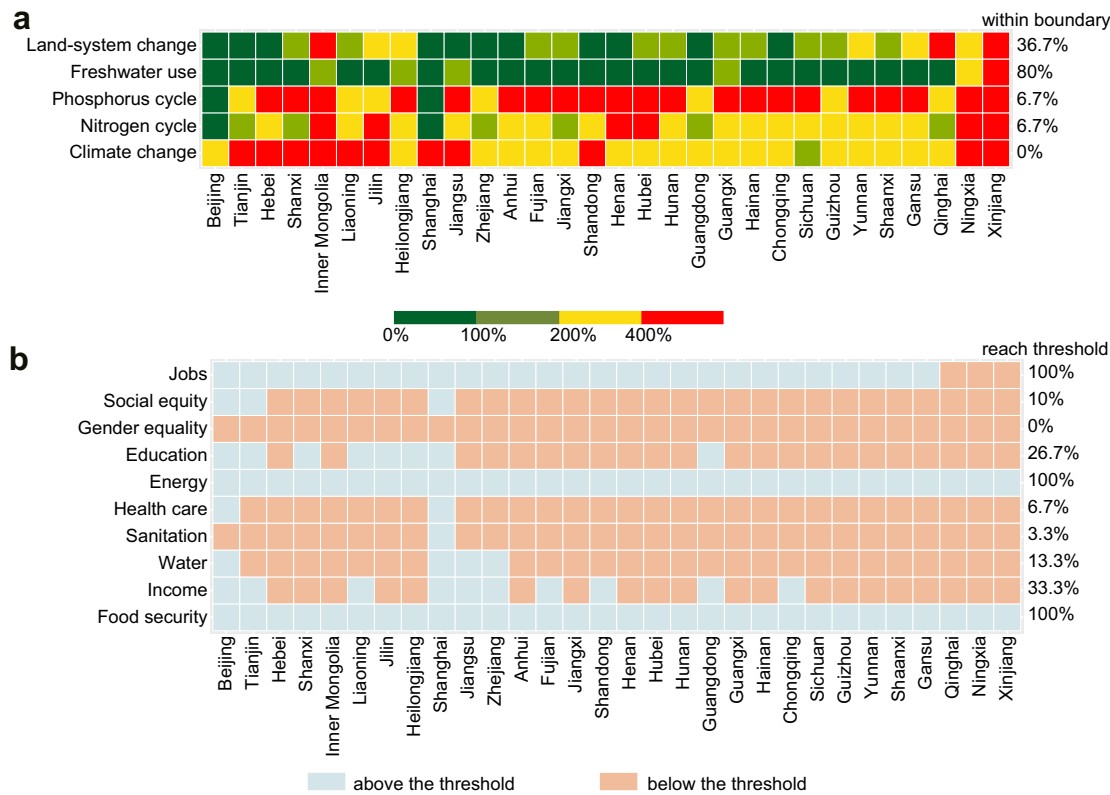

**Fig. 4 | Sustainability performance of China's provinces from 2000 to 2018. a** Environmental performance with respect to per capita environmental boundaries. **b** Human well-being with respect to social foundations.

identified the threshold value based on the targets of the SDGs (Fig. 4b and Fig. 5b). Human well-being can be measured by the ratio of actual social indicators to social thresholds.

For the spatial pattern of human well-being in China, the results are quite complex. At the provincial scale (Fig. 4b), China's provinces perform overall well on food security, energy, and jobs, with all provinces reaching the thresholds. Close to 1/3 of the provinces achieve the threshold of household income. The number of provinces that achieve the foundations for education, health care, social equity, water, and sanitation is respectively 26.7%, 6.7%, 10%, 13.3%, and 3.3%, respectively. In contrast, provinces underperform on gender equality, with no provinces reaching this threshold. Notably, no province achieves all 10 social thresholds. The numbers of social foundations reached across provinces range from two to nine (i.e., Shanghai).

For the temporal changes in social performance, an overall improvement in human well-being has been observed in China from 2000 to 2018, with developments in all aspects (Table S14). At the provincial scale (Fig. 5b), 5 out of 10 social indicators of all provinces have increased (i.e., water, sanitation, health care, education, and gender equality) during 2000–2018. In addition, income and energy have increased in most provinces, except for those with no trend of change. 90% of the provinces have increased in food security, except Tianjin, Heilongjiang, and Guangdong. Jobs have increased in 93% of the provinces, except Shanxi and Shandong. In contrast, 73% of the provinces have decreased in social equity. Increased changes are mainly distributed in western provinces.

Overall, our results revealed marked spatial heterogeneity and temporal dynamics in sustainability performance that are specific indicator-dependent. These results evidence that monitoring only environmental or social performance might be insufficient when attempting to measure progress towards sustainability.

## Assessment of sustainability performance based on coupling interaction

Based on our analysis of sustainability performance on environmental performance and human well-being (i.e., the quantification of SJOS), we further calculated the coupling coordination degree, to quantitatively measure the strength and direction of the interactions (e.g., synergies or trade-offs) between environmental performance and human well-being.

We analyzed the spatial and temporal variations of human-environment interactions based on CCD for all of the provinces in China over the period 2000–2018 (Fig. 6). High values depict synergies (coupling), while low values depict trade-offs (decoupling) between environmental performance and human well-being. To further compare the relative performance of biophysical and human subsystems, we distinguished system into: environmental development lag (i.e., environmental performance below human well-being), social development lag (i.e., environmental performance above human well-being), or environmental-social synchronization (i.e., environmental performance matches well with human well-being) types. To support the analysis and facilitate interpretation of the results, we categorized CCD results into five levels (Fig. 6a) and distinguished respective social/environmental lag (Table S16).

For the spatial variations of CCD, China is overall in moderate coordination with social lag stage (Table S16). At the provincial scale (Fig. 6a), our analysis revealed significant heterogeneity in CCD across provinces (ranging from 0.075 to 0.993). Our results showed that the spatial pattern of CCD seems to increase from the western regions to eastern regions. High values are concentrated in the eastern region, and low values are mostly located in the central and western regions. Specifically, strong synergies, namely, high coordination are mainly located in eastern China. These regions have strong synergies between

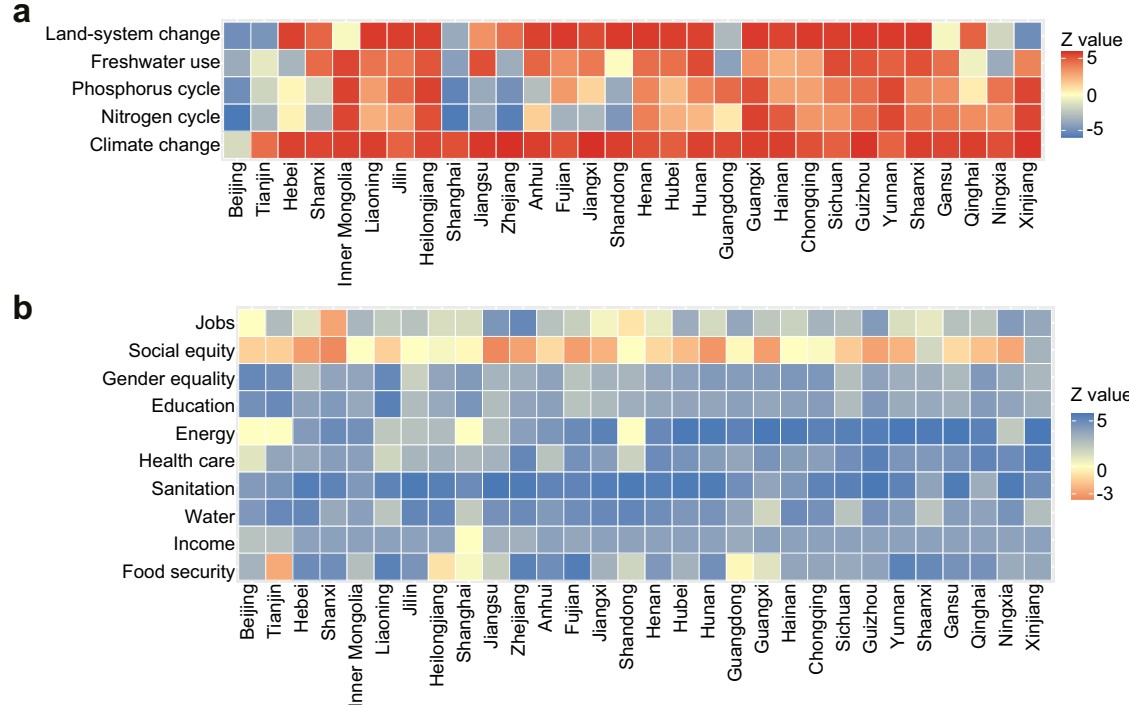

**Fig. 5 | Changing trends of sustainability performance in China's provinces from 2000 to 2018. a** Environmental performance. **b** Human well-being. Environmental performance indicates environmental footprints to downscaled planetary boundaries. Human well-being represents social indicators to social thresholds. Z-score values indicate the results of Mann-Kendall test, with red color representing provinces with negative changes and blue color representing provinces with positive changes. Specifically, negative changes in environmental performance and human well-being represent increase in footprints and decrease in social indicators, respectively. The given significance level α is 0.05.

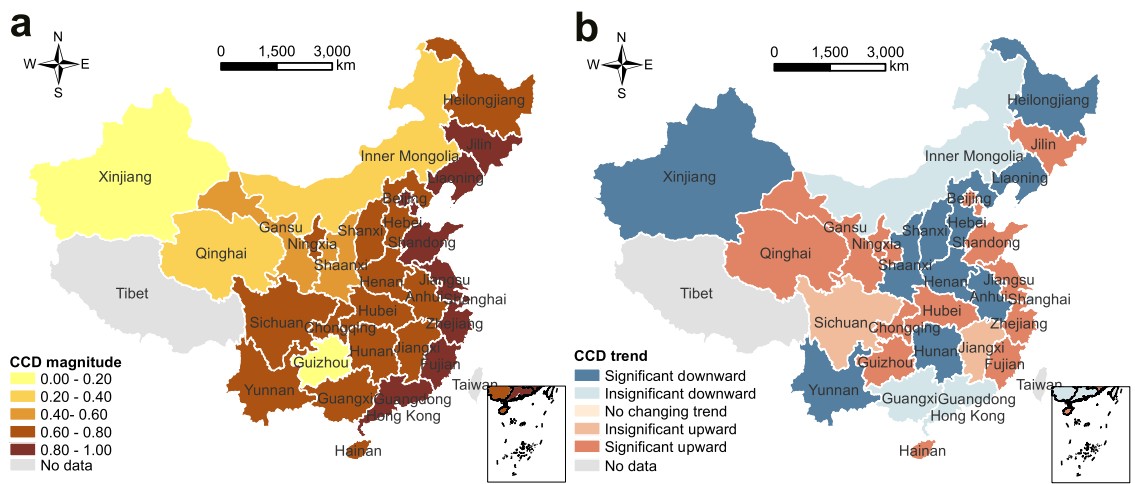

**Fig. 6 | Spatial patterns of coupling coordination degree between environmental performance and human well-being from 2000 to 2018. a** The magnitude. **b** Changing trends. The coupling coordination degree scopes are listed in Table S5, referring to Shi et al.[36] and Li et al.[78]. The given significance level α is 0.05.

environmental performance and human well-being. Moderate coordination is mainly located in central China. Primary coordination, intermediate unbalanced, and extreme unbalanced are located in western China, with strong trade-offs between environmental performance and human well-being.

To further understand the relative lagging aspects, which hinder the coupling coordination development, we compared the provinces' performance on environmental and socio-economic dimension (Table S16). Overall, 21 out of 30 provinces are social development lag type. That means development of human well-being lags behind environmental performance and hinders sustainable development in these regions. Environmental development lag type is located in western China (Xinjiang, Inner Mongolia), which are economically underdeveloped regions. In contrast, socio-environmental synchronization type is mainly in the eastern regions, which is the most balanced situation in terms of environmental and socio-economic development. Notably, high-coupling coordination regions are mainly located in the eastern plains, while low-coupling coordination regions are in the western areas, which is consistent with the spatial distribution of social and environmental performance. This result indicated that coupling interactions among subsystems match well with the environmental and social performance. Namely, where CCD is high, it is mostly socio-

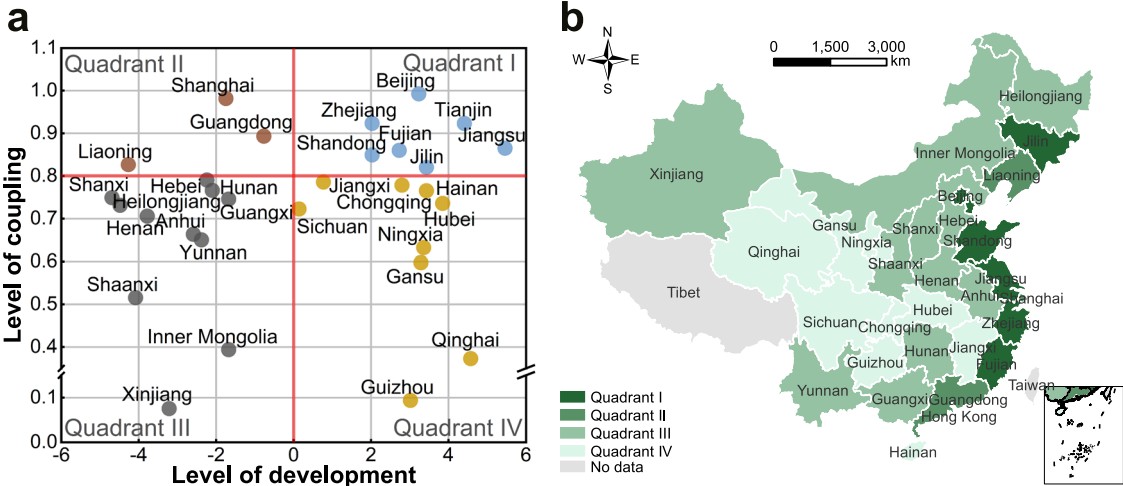

**Fig. 7 | Development patterns of China's provinces from 2000 to 2018. a** The performance relative to the levels of coupling (*y*-axis) and development (*x*-axis). **b** Spatial patterns. The level of coupling represents the magnitude of CCD (i.e., from Fig. 6a). The level of development represents the changing trends of CCD (i.e., from Fig. 6b). Corresponding to Fig. 2, Quadrant I, Quadrant II, Quadrant III, and Quadrant IV represent coupled and developed, coupled and underdeveloped, uncoupled and underdeveloped, and uncoupled and developed types, respectively.

environmental synchronization; where CCD is low, it is mostly social or environmental development lag. These results, therefore, suggest that improving either environmental performance or social well-being does not necessarily mean transitioning toward sustainability.

To trace the temporal changes, we applied the Mann-Kendall trend test and Sen's slope estimator to analyze the trends of CCD from 2000 to 2018. At the national scale, CCD shows an insignificant upward trend over time at the national scale (Table S16). At the provincial scale (Fig. 6b), CCD has increased in 16 of the 30 provinces from 2000 to 2018. Specifically, 14 out of 30 provinces with significant progress mainly located in western and eastern regions. Our analysis revealed that CCD for most provinces have increased over this period, indicating progress toward achieving sustainability in China.

To elaborate on the drivers of coupling interactions, we quantified the contributions of 24 factors of three categories to changes in CCD, each of which could potentially promote or hinder the synergistic development between environmental and socio-economic development goals (i.e., magnitude of CCD). This analysis allows us to understand the major driving forces behind them, to ultimately assess the level of urgency in tackling incoordination and how to address it more effectively. To this end, we developed an empirical diagnostic model based on boosted regression trees, a machine-learning technique extended from traditional classification and regression trees. The boosted regression tree models successfully explain more than 90% of the changes of CCD in all indicators.

According to our analysis, differences in coupling coordination relationships for climate change, freshwater use, land-system change, phosphorus cycle, nitrogen cycle, and overall environmental performance are all mainly attributed to environmental factors (Fig. S18). Yet key drivers differed by each indicator (Fig. S19). The dominant drivers for climate change, land-system change, and overall environmental performance are grass area, accounting for 26%, 22%, and 45% of all drivers, respectively (Fig. S19a, c, f and Table S17). On the other hand, key driver for nitrogen and phosphorus cycle is the urbanization rate, which contributes 22% and 21% of all factors (Fig. S19d, e and Table S17), respectively. For freshwater use, CCD is mostly influenced by Normalized Difference Vegetation Index (NDVI), accounting for 32% among all 24 factors (Fig. S19b and Table S17).

To further understand how these main drivers affect changes in CCD, we examined the influence of the dominant drivers for each indicator. For climate change, land-system change, and overall environmental performance, our results showed that the overall influence

decreases with the increase of grassland area. By referring to the partial dependence on the driving factors (Fig. S20), the relative impact of grassland area is mainly positive, indicating that the increase in grasslands promotes coupling coordination. For nitrogen and phosphorus cycle, the overall influence increases with the increase of urbanization rate. The urbanization rate contributes to the decrease of CCD. For freshwater use, the overall influence increases with the increase of NDVI. From the decile scale, it can be seen that only 10% of the data are in this range of 0–0.4. These results suggest that NDVI promotes synergistic development between freshwater use and human well-being.

These results, therefore, highlight the need to consider the environmental and socio-economic dimensions of sustainability simultaneously to help identify and address the trade-offs between them and establish realistic expectations on how best to meet human basic needs in a sustainable manner.

## Towards regional sustainability: development patterns and strategies

As mentioned above, our results emphasized the need for targeted policies and strategies to promote environmental and socio-economic synergies towards regional sustainability. To contribute to effective policy-making, we proposed a roadmap based on a two-step approach (Fig. 2). In the first step, different development patterns are delineated considering CCD magnitude (i.e., level of coupling in Fig. 6a) and CCD trends (i.e., level of development in Fig. 6b). In the second step, targeted development strategies are recommended based on the characteristics of each pattern and their underlying drivers or causes.

Based on the performance relative to the levels of coupling and development, China's provinces are divided into four categories: coupled and developed, coupled and underdeveloped, uncoupled and underdeveloped, and uncoupled and developed types (Fig. 7a). For China's provinces, 7 out of 30 provinces are in Quadrant I, with relatively ideal development pattern, mainly located in eastern China. These regions perform well both in coupling and development aspects, in a high coordination phase and moving toward more coupling simultaneously. Three provinces in Quadrant II are coupled and underdeveloped, located in eastern region. Quadrant III represents the provinces, uncoupled and underdeveloped. Eleven provinces are located in this quadrant, mainly located in central and western regions. Finally, nine provinces are in Quadrant IV, with uncoupled and developed status. These provinces are mainly located in western China. As

indicated by our results (Fig. 7b), the western regions have the lowest level of coupling and the least improvement in coupling, suggesting that the government needs to pay more attention to this uncoupled and underdeveloped pattern. In contrast, the eastern regions have the highest levels of coupling and development. This could be attributed to the higher investments to protect the environment and reduce the environmental footprints at the more economically developed stage.

Based on performance characteristics and main drivers of each development pattern, we further elaborated on tailored policies (see Discussion) for each Quadrant of provinces so as to bring and keep both environmental burdens and social well-being distribution within the desired range and in a sustainable manner. In this context, our approach could help to establish more effective targets and make better-informed policy decisions.

## Discussion

### Implications of SJOS towards sustainability

Our research integrated environmental performance with human well-being and their coupling coordination interactions to assess sustainability performance and quantified their spatial and temporal variations relative to the defined SJOS.

Sustainability performance on environmental and socio-economic dimensions within China varies substantially from region to region and has changed obviously over time. In general, provinces in eastern China tend to operate within more environmental limits and reach more social foundations, such as Shanghai, Beijing, and Tianjin (Fig. S15). This spatial heterogeneity may result from regional divergence, such as (a) heterogenetic conditions of climate, terrain, soil, and natural resources, (b) population density, (c) industry structure (agriculture, light industry, and heavy industry), and (d) policy implementation, which have their own specific effects[46]. Hence, our results revealed significant regional disparities in sustainability performance relative to SJOS, including environmental and socio-economic dimensions, highlighting the need to further examine the linkages and coupling relationships between environmental performance and human well-being and underlying causes for spatial-temporal variations to achieve regional sustainability.

The socio-economic and environmental dimensions are inextricably linked and collectively influence sustainability performance. In addition to understanding sustainability performance relative to SJOS, our work also focused on the extent of coupling coordination relationships between environmental performance and social achievements. CCD results showed an overall moderate coordination as lagged social development at the national scale. In general, the eastern region (see Fig. S3) has a higher level of coordination than the western and central regions (Fig. 6a), with significant progress over the period 2000–2018. These spatial-temporal patterns are external manifestation of several underlying mechanisms affecting coupling coordination relationships. Our driver analysis demonstrated that coupling coordination relationships are mainly influenced by environmental factors. Land use/cover pattern (i.e., grassland area) and urbanization rate are the dominant drivers of changes in CCD between environmental performance and human well-being. In specific, the main drivers are NVDI for freshwater use, grassland area for climate change and land-system change, and urbanization rate for nitrogen and phosphorus cycles. Based on our analysis, grassland area and NDVI generally result in an increase in CCD. This implies that increasing vegetation cover (i.e., grasslands and forests) may contribute to sustainable development, reducing the imbalance between ecological protection and socio-economic development[47]. This could be attributed to the implementation of an integrated portfolio of large-scale sustainability interventions in response to ecosystem degradation from rapid economic development, including ecological programmes[41] and investments in natural capital[48], particularly the Grain for Green Program and the Natural Forest Conservation Program. In response, China's forest cover has

transitioned in recent decades, turning from net loss to gain[49,50]. Grassland ecosystems in northern and western China have responded to large-scale restoration and grazing exclusion, with increasing grasslands via conversion from deserted land and low-yield cropland[41]. These interventions have slowed down deforestation, promoted ecological restoration, and enhanced ecological conditions. Studies have shown that China's ecosystem services have improved over the past 20 years, sustaining and enhancing human well-being[48]. The synergistic development of environmental and socio-economic dimensions improves the CCD. In contrast, urbanization rate has led to a decrease in CCD. With rapid urbanization, 55% of the world's population lives in urban areas in 2018 and could reach 68% by 2050[51]. Approximately 80% of global gross domestic product (GDP) is generated in cities[52]. Whereas city residents are important contributors to environmental degradation, e.g., being responsible for about 80% of global greenhouse gas emissions[53]. Cities discharge large amounts of wastewater containing N and P elements. Since the late 1970s, China has faced great environment concerns resulting from rapid urbanization development, particularly the pollution of its air, water, and soil[54,55]. These emissions are particularly worrisome when they exceed the Earth System's carrying capacity, resulting in environment lags. Imbalance between environment and socio-economics may lead the decrease in CCD. It should be noted that China has made tremendous progress in urban environment governance over the past decades[56], although this task remains intractable. Hence, the achievement of sustainable development goals to PBs is largely determined by cities, as they drive cultures, economies, material use, and waste generation[13]. To achieve sustainable development, we must adhere to a new development paradigm: harnessing the growth and development benefits of urbanization while actively managing its negative environmental impacts.

### Policy recommendations towards promoting regional sustainable development

Improvements in environmental performance or human well-being cannot guarantee that the regions transition towards sustainable development. Our results for coupling coordination relationships exhibit spatial heterogeneity, and in most regions, show trade-offs between environmental and social-economic aspects. However, opportunities exist to mitigate such trade-offs through targeted sustainable management strategies. Therefore, strategies will need to be imposed by region considering coupling coordination development between human well-being and environmental performance to ultimately succeed in mitigating trade-offs and realizing sustainability.

**Synergistic strategy.** For coupled and developed pattern in Quadrant I (Fig. 7), the level of coupling and development is relatively high. These regions are with synergies between human well-being and environmental performance. Among these regions, Beijing, Tianjin, Jiangsu, Zhejiang, Fujian, and Shandong, located in the coastal areas (see Fig. S3), have a high degree of decoupling between economic development and resource consumption[57]. These provinces are among the most developed areas in China (e.g., high GDP and household income) with rapid technological progress, adequate human and social capital, and a large influx of high educated populations. One reason for this pattern in east-coastal areas is related to policy implementation[32]. At the beginning of the Chinese reform and opening-up policy, the Chinese government focused on facilitating economic development more in east-coastal areas[58]. In addition, eastern China has a relatively flat terrain (majority of terrain is plain), making it more favorable for transportation[59] and its climate conditions (e.g., precipitation)[60]. Therefore, it is recommended that these regions take more responsibility for interregional cooperation by providing access to human and financial resources and new technologies[57]. In general, these provinces represent the forefront of sustainable development in China and can serve as a typical pattern for other regions to achieve more resilient

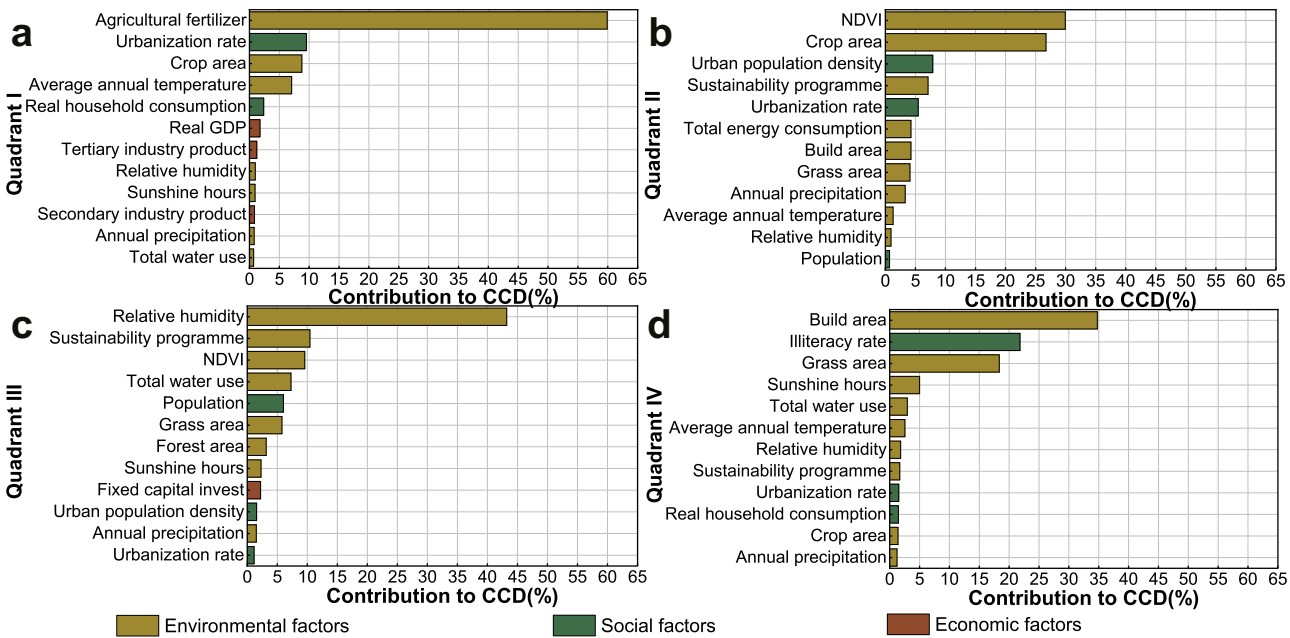

**Fig. 8 | Contribution of drivers to variations of coupling coordination degree between environmental performance and human well-being. a** Quadrant I. **b** Quadrant II. **c** Quadrant III. **d** Quadrant IV. Shown are the relative contributions, as integer percentage (%). For each indicator, we show the contribution of each of the three factors towards the changes in coupling coordination degree: environmental (brown), social (dark green), and economic (wine red) factors. All the drivers in each development pattern can explain at least 90% of the changes in coupling coordination degree from 2000 to 2018.

and sustainable development, especially for some developed nations or regions. In this pattern, current development strategies can maintain as usual, while encouraging further improvements. For example, agricultural fertilizer is the main driver in this pattern, generally resulting in a decrease in CCD (Fig. 8a and Fig. S21a). Thus, environmental issues should receive more attention.

**Promotion strategy.** For coupled and underdeveloped pattern in Quadrant II (Fig. 7), the level of coupling is above the dividing point, yet tends to become lower. Regions in this pattern are in high coordination with a decrease of CCD. NDVI is the main driver in this pattern (Fig. 8b). An increase of NDVI generally indicates a decrease in CCD (Fig. S21b). This is due to the uneven development of this pattern, for example, some areas have a good ecological environment, while the economic level is relatively backward. This suggests that current policy of prioritizing ecological conservation has not accompanied by improvement in human well-being, resulting in an imbalance between environmental and socio-economic development. It is recommended to prioritize policy coherence across environmental, social, and economic goals[61]. For resource-based regions, such as Liaoning, the improvement of socio-economic development is mainly hindered by population loss, low fertility, and an aging population[60]. To response to this underdeveloped pattern, promotion strategy should be implemented. Promotion strategy should aim to increase investment and allocation of resources in lagging aspects to provide substantial support[44]. For this strategy, sufficient funds should be provided to guarantee the construction of infrastructure to improve the living environment and life quality of residents[62]. With sufficient economic support, these regions are able to promote harmonious environmental and socio-economic development. In addition, encourage policies should be implemented to attract talent and discourage regional population loss.

**Transformative strategy.** For uncoupled and underdeveloped pattern in Quadrant III (Fig. 7), the levels of development and coupling are both below the dividing points (Fig. 7a). Among these regions, most provinces are rich in fossil energy and mineral resources but lagging environmental performance (Table S18), especially in western China

(Fig. 7b). For example, some provinces benefit primarily from economic development in the energy sector, such as Inner Mongolia and Shanxi with reserves of coal resources. Since these areas are located in China's arid and semi-arid regions, the main driver affecting CCD in this pattern is relative humidity (Fig. 8c and Fig. S21c). Water scarcity is a major constraint affecting the development pattern in these regions. As one of the main sources of electricity production and the energy industry, Inner Mongolia holds a considerable proportion of energy consumption and carbon emissions in China[57]. As energy exporters, these provinces experience much more stress because they share the environmental responsibility of energy importers (e.g., substantial emissions and resource consumptions)[63,64]. In addition, economic backwardness in these regions hinders technological upgrading and thus weakens their capacity to improve energy efficiency. Growing resource intensity and fossil fuel dependency along with slow technological change pose high challenges to sustainable development, leading to imbalance between environmental performance and human well-being. Thus, the problem in this pattern is how to develop efficient technologies for energy production, reducing dependence on resource throughput and mitigate environmental costs. Transformative strategy should aim to decouple ecological degradation from the increase in social outcomes. For this strategy, transitions in industrial structure (i.e., development in tertiary industry) and energy mix can be a possible pathway to reduce the negative impact from energy sector. In addition, regional cooperation and technological support from developed regions can be boosted to promote technological updates in reducing fossil fuel consumption.

**Fill-gap strategy.** For uncoupled and developed pattern in Quadrant IV (Fig. 7), the level of coupling is below the dividing point, yet has improved from 2000 to 2018 (Fig. 7a). In this pattern, most regions are less-developed regions in the northwest (Fig. 7b), especially Gansu, Qinghai, and Ningxia. These provinces located in ecologically fragile areas with low ecological carrying capacity, and therefore environmental factors remain the main drivers of this pattern. Build area is the main driver shaping this pattern, which has generally led to a decrease in CCD (Fig. 8d and Fig. S21d). Economic development in this pattern is

highly dependent on carbon-intense industries (e.g., construction industry)[57], thus causes serious damage to the environment. Traditional development patterns can exacerbate the environmental vulnerability of these regions, thus leading to low level of coupling between environmental performance and human well-being. Fill-gap strategy can guide management interventions to address the cause of trade-offs to maintain and enhance progress towards a more coupled direction. The recommended solution for this pattern is to promote the transformation of energy-intensive industries and non-fossil fuel development. These regions have great potential to develop clean energies (such as solar energy, wind energy, and hydropower), which can completely meet increasing energy demand as well as maintain a sustainable environment for future generations. Thus, these regions should pay more attention to develop efficient clean energy utilization technology.

## Opportunities for localizing sustainability

To better integrate our framework with management actions and environmental governance, our future research priorities aim to connect the global SDGs with locally specific actions, which require better consideration of the local environmental, contextualized socioeconomic factors, and aspiration and interests of diverse stakeholders (e.g., local communities, small-scale business, cities, etc.)[65]. As proposed by Moallemi et al.[65], the jointed framing of context-specific goals via genuine stakeholder engagement from the bottom-up processes can complement our framework's guidance for policy-making. Stakeholder engagement through participatory processes can help define and regulate the local SJOS, including setting environmental limits, social thresholds, and judging whether proposed policies are actionable and feasible.

Second, to better apply our conceptual framework in localizing sustainability, we can attempt to define the environmental limits at local scales using a harmonized approach to local use of PBs that combines the advantage of the fair shares approach (Earth System relevance and global responsibility) and the local safe operating space approach (local relevance). Using this allocating approach can ensure that actions in a local scale contribute to sustainability at all scales, from local to global[66].

## Research limitations

A limitation of this research pertains to the selection of control variables and corresponding indicators. For example, the PB-based sustainability assessment only focused on a subset of deliberately selected environmental indicators[26]. We adopted the same set of indicators so that we can compare and place our results in a global context. However, we also acknowledge that other indicators (e.g., those related to air and water pollution and resource use) can also be critical for regional sustainability and have attempted to incorporate additional metrics for assessment related to environmental quality, in reference to key aspects of national policy concerns (Figs. S9–S12), to link regional and global sustainability realistically and effectively[5]. Furthermore, to clarify the responsibility of different stakeholders for sustainable development, the PB-based sustainability assessment should be more holistic and inclusive, and reflect contributions from different perspectives (i.e., differences between production-based or consumption-based methods)[67]. The jointly framing of multiple sustainability perspectives (i.e., consideration of both production and consumption-based estimates) can more comprehensively assess and reflect the actual human-induced environmental pressures. To this end, we have further attempted to compare the consumptive and territorial performance (Figs. S4–S8). There are disparate quantitative results between production and consumption-based perspectives, yet these are not sufficient to alter the category (i.e., safe/increasing risk/high risk) of most results in terms of environmental performance.

Another limitation is related to the inherent uncertainty of the PB framework (Table S4) and the downscaling method, which could

influence our main results. For example, we adopted the fair share principles based on population size and per capita values, which is the most common downscaling approach[29]. Whereas the scientific community has not yet reached an agreement on how to allocate shares since this issue is fundamentally policy-oriented and can have ethical implications. Nevertheless, the PB results can be sensitive to the choice of particular downscaling methods. To this end, we did investigate the sensitivity of coupling coordination relationships to changes in fair share method, showing that downscaling method has the greatest impact on the results related to climate change (Fig. S22).

We acknowledge that these are methodological concerns due to sharing principles, control variables and selected indicators, and sustainability perspectives, but offers avenues for future research. For example, setting environmental boundaries inevitably involves value judgments that should resonate with practitioners or stakeholders, which can be achieved through transdisciplinary research and knowledge coproduction[68]. This needs to be further addressed through comparing a range of different sharing principles, downscaling approaches, and sustainability perspectives. Moreover, future research needs to explore additional dimensions based on the specific study region and local context to refine the PB-based sustainability assessment framework, such as inclusion of other indicators pertaining to environmental quality, critical resource use natural capital.

## Methods

### Environmental limits and environmental performance

To measure the environmental performance and compare it with the corresponding environmental limits, the original control variables of the planetary boundaries need to be translated from state to pressure[26], which allows for effective monitoring by governments and other actors. The majority of the original PBs are conceived as aggregate effects from locally heterogeneous environmental states or pressures, we adopted a top-down downscaling to define environmental limits that follow an equal share per capita approach. In the equal rights per capita approach, people are selected as the direct beneficiaries of the allocation[69]. For aggregated processes, these boundaries can be relatively straight-forward allocated according to total annual population. For systemic processes, this global limit per capita is computed differently for yearly budgets. For indicators considered as yearly budgets (climate change), computing an equal share per capita value requires considering current and future populations of the Earth. To downscale this boundary, we need to divide budget by the sum of all yearly inhabitants until 2100. The limit per capita evolves each year according to the yearly global population, using the interpolated total population data in medium fertility until 2100.

In our study, we downscaled four planetary boundaries (climate change, biogeochemical flows, freshwater use, land-system change) to per capita equivalents, reference to the approach revised from Dao et al.[69] and Algunaibet et al.[70]. As two PBs are defined for biogeochemical flows (nitrogen and phosphorus cycles), five environmental indicators are considered. Per capita environmental limits are then compared to the corresponding environmental footprints (carbon footprint, nitrogen footprint, phosphorous footprint, blue water footprint, and land footprint). The data sources for environmental footprints are listed in Table S6. Environmental performance is a quantitative score computed as the ratio of a footprint over a limit. Environmental performance is classified into three categories: safe, increasing risk, and high risk, shown in Table. S3.

### Climate change

The original boundary has been set at a maximum 350 ppm concentration of atmospheric $CO_2$, or 1 watt per m² of additional radiative forcing compared to pre-industrial levels, which should keep global warming below 2 °C[19]. The global limit for climate change is set with the remaining cumulative $CO_2$ emissions for a "medium" probability

(50%) to stay below a 2 °C increase by 2100 compared with pre-industrial level[71]. The sum of inhabitants from 2018 to 2100 is 814.44 billion people-year[72]. Equal per capita allocations to all inhabitants of climate change would translate to allowable annual $CO_2$ emissions of 1.54 tons per capita from 2000 to 2018.

## Freshwater use
An equal per capita allocation of the original planetary boundary (maximum total consumptive blue water use of 4000 km³ per year, according to Rockström et al.[19] would translate to an allowable annual blue water use of 587 m³ per capita per year.

## Land-system change
An equal per capita allocation of land use according to the original planetary boundary[19] would translate into per capita anthropogenic land use of 0.29 hectares per capita per year, or alternatively limiting agricultural and urbanized area to 15% of ice-free land[29].

## Biogeochemical flows
The original boundary of nitrogen cycle is 62 Tg/year nitrogen, including intended biological and chemical N fixation[20]. According to Steffen et al.[20], the current value of N flow is 150 Tg N per annum, out of which 96 Tg N per annum (64%) is attributed to chemical fixation by fertilizers. Referring to Algunaibet et al.[70], we reduced the N cycle planetary boundary from 62 to 39.7 Tg N per annum to consider industrial fixation only, assuming such share would remain constant. An equal per capita allocation of the planetary N boundary (39.7 Tg N y-1 from industrial fixation) would translate to about 5.82 kg N per capita per year during the period of 2000–2018. The original boundary for phosphorus cycle (6.2 Tg P y-1 mined and applied to erodible agricultural soils)[20] would translate to about 0.91 kg P per capita per year.

## Additional regional environmental indicators
Besides environmental limits calculated by PB indicators, we also considered additional indicators from a regional context perspective related to environmental quality, in reference to key aspects of national policy concerns. Supplementary assessment (based on data availability) includes indicators pertaining to air quality, water quality, and resource use (Figs. S9–S12).

## Social thresholds and human well-being
We selected 10 social indicators following SJOS framework and social objectives contained in the SDGs[22]. The data sources for these social indicators are listed in Table S7. Raworth et al.[22] identifies 11 social indicators to guarantee human rights and corresponding foundations in the Rio+ 20 conference. The SDGs identify 17 goals, of which 12 can be categorized as social objectives. These goals relate directly to the satisfaction of basic human needs (human well-being), such as ending extreme poverty, ending hunger and malnutrition, assess to clean water and sanitation, and access to affordable and clean energy. Other goals correspond indirectly with humanity's impact on the environment (environmental footprints), such as responsible consumption and production[73]. In general, the goals fit fairly well with the social foundations in the SJOS framework. The social thresholds reference to the SDGs indicators and the thresholds in O'Neill et al.[30]. Human well-being is a quantitative score computed as the ratio of a social indicator over a threshold. The social foundation is considered to be achieved when the ratio reaches the threshold.

## Mann-kendall trend test and Sen's slope estimator
Mann-Kendall method[74,75] is applied to measure the long-term changing trends in the environmental footprints, social indicators, environmental performance, human well-being, and coupling coordination degrees. This non-parametric method does not specify whether the trend is linear or nonlinear. This approach is robust for non-normally

distributed data and has low sensitivity to outliers. Thus, it has been widely applied to detect the significance of trends in time series.

In the Mann-Kendall test, with the null hypothesis $H_0$, the time series data $(x_1..., x_n)$ include a sample of $n$ independent and random variables with the same distribution. With the alternative hypothesis $H_1$, there is an increasing or decreasing trend in the time series. The statistic $S$ is defined as follows:

$$S = \sum_{i=1}^{n-1} \sum_{j=i+1}^{n} sgn(x_j - x_i) \tag{1}$$

$$sgn\left(x_j - x_i\right) = \begin{cases} +1, x_j - x_i > 0 \\ 0, x_j - x_i = 0 \\ -1, x_j - x_i < 0 \end{cases} \tag{2}$$

where the time series length $n = 19$, $x_i$ and $x_j$ are the data values in time series $i$ and $j$ $(j > i)$, respectively. The test is conducted using the $Z$ value:

$$Z = \begin{cases} \frac{S-1}{\sqrt{VAR(S)}}, S > 0 \\ 0, S = 0 \\ \frac{S+1}{\sqrt{VAR(S)}}, S < 0 \end{cases} \tag{3}$$

$$VAR(S) = (n(n-1)(2n+5) - \sum_{i=1}^{m} t_i(t_i - 1)(2t_i + 5))/18 \tag{4}$$

where $n$ is the number of data points, $m$ is the number of nodes in the time series, and $t_i$ is the width of the node. Positive values of $Z$ indicate increasing trends while negative $Z$ values indicate decreasing trends. The null hypothesis is accepted when $|Z| \leq Z_{1-\alpha/2}$, and the trends within the time series is considered to be insignificant. When $|Z| > Z_{1-\alpha/2}$, the null hypothesis is rejected, and a significant trend exists in the time series. At the given significance level of $\alpha=0.05$, the null hypothesis of no trend is rejected if $|Z| > 1.96$.

Sen's slope[76] is a nonparametric procedure for estimating the slope of trend. We used Sen's slope method to measure the changing magnitude of time series from 2000 to 2018:

$$\beta = Median\left(\frac{x_j - x_i}{j - i}\right) \tag{5}$$

## Data normalized method
To eliminate the magnitude and measurement of different data, we further standardized the data of environmental performance and human well-being by adopting the maximum difference normalization model as follows:

$$Positive\ indicator : Y_{ij} = \frac{X_{ij} - X_{minj}}{X_{maxj} - X_{minj}} \tag{6}$$

$$Negative\ indicator : Y_{ij} = \frac{X_{maxj} - X_{ij}}{X_{maxj} - X_{minj}} \tag{7}$$

where $Y_{ij}$ refers to the normalized values; $X_{ij}$ refers to the origin values of $j_{th}$ indicator in year $i$; $X_{maxj}$ and $X_{minj}$ are the maximum and minimum values of $j_{th}$ indicator, respectively.

## Coupling coordination degree model
The coupling theory is an effective method to test the relationship between two or more systems that have interactions with each other. In this paper, it is used to investigate the coupling and coordination

interactions between environmental performance and human well-being in a SJOS. In detail, we first calculated the coupling degree in formula (8), then measure the coupling coordination degree in formulas (9) and (10). The formulas are as follows, referring to Yang et al. (2020)[77]:

$$C = \sqrt{\frac{f_{(X)} \times f_{(Y)}}{\left(\left[f_{(X)} + f_{(Y)}\right]/2\right)^2}} \qquad (8)$$

where $C$ refers to the coupling degree, $C \in [0, 1]$. The greater the coupling degree is, the stronger interaction between the subsystems would be, and vice versa; $f_{(X)}$ and $f_{(Y)}$ represent the environmental performance and human well-being, respectively.

$$D = \sqrt{C \times T} \qquad (9)$$

$$T = \alpha f_{(X)} + \beta f_{(Y)} \qquad (10)$$

where $D$ represents the coupling coordination degree, $D \in [0, 1]$. Higher values of coupling coordination degree represent synergies between environmental and socio-economic systems. $T$ refers to comprehensive development level. $f_{(X)}$ and $f_{(Y)}$ refer to environmental performance and human well-being. Environmental performance indicates the ratio of environmental footprints to environmental limits (downscaled planetary boundaries). Human well-being indicates the ratio of social indicators to social thresholds. $\alpha$ and $\beta$ are the weights indicating the importance of each subsystem, respectively, and $\alpha + \beta = 1$. In our study, we assume that the biophysical and natural system is equally important within a social-ecological system. Thus, $\alpha = \beta = 0.5$ is set.

Referring to the division of coordination types in physics, the coupling types of environmental and social performance are split according to the classification criteria of coupling coordination degree given by Shi et al.[36] and Li et al.[78]. By comparing the environmental performance and human well-being, we further divided coupling coordination degree into three types: environmental development lag type, social development lag type, environmental-social synchronization type. Different ranges of values represent different correlations between environmental and socio-economic aspects (Table S5).

### Boosted regression tree method

To understand the spatial and temporal variation mechanisms of coupling coordination degrees, a boosted regression tree approach is adopted to analyze the relative contributions of driving factors to the spatial and temporal variations of coupling coordination degree. The boosted regression tree method is a machine learning technique extended from traditional classification and regression trees, which combines the algorithms of regression trees that use recursive binary splits to fit a simple model to each result and boosting that uses an iterative method to gradually add trees to develop the final model[79]. Compared to the commonly used multiple linear stepwise regression, the BRT method can fit complex nonlinear relationships and automatically handle interaction effects between predictors.

In our models, the spatial and temporal changes in coupling coordination from 2000 to 2018 are response variables, and the 24 driving factors (i.e., predictor variables, listed in Table S17) are predictor variables. Driving forces are selected from socio-economic-environmental aspects, mainly with reference to dimensions framed by shared socioeconomic pathways[80], such as demographic and human development, economy and lifestyle and policies and institutions, and technology and environment and natural resources elements. The specific factors are mainly chosen based on the results of the literature analysis of environmental footprint drivers[81–85] and the availability of data. The data sources for these drivers are listed in

Table S8. Three parameters in our study are specified, including gaussian error distribution, a learning rate of 0.001, an interaction depth of 5, and a bag fraction of 0.5, as recommended[79,86]. All analyses are conducted with R (version 3.4.3), modeling with "gbm" package plus custom code that is available online[79].

### Reporting summary

Further information on research design is available in the Nature Portfolio Reporting Summary linked to this article.

## Data availability

Our research relies on data from multiple sources, all sources for environmental and social indicators are listed in Tables S6 and S7. All sources for driving factors are listed in Table S8. All data are for the year from 2000 to 2018. The global data used for comparison are mainly from EDGAR[87], FAOSTAT[88], World Bank[89] and Eora MRIO[90,91] databases. World population is from UNPD[72]. China's provincial data are from CEADS database (www.ceads.net/data/), Resource and Environment Science and Data Center (https://www.resdc.cn/) and China Statistical Yearbooks. Additional environmental data for China are from ACAG[92] database and China Statistical Yearbooks. All the data generated in this study are provided in the Supplementary Information/Source Data file. Source data are provided with this paper.

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

## Acknowledgements

This study is supported by the programs of the Second National Qinghai-Tibet Scientific Research Project: Climate Change in Plateau Regions with Scarce Climate Data and Its Impact and Response (Grant No. 2019QZKK1001, Y.D.), the National Natural Science Foundation of China (41971269), the Key Project of Science and Technology Department of Qinghai Province (Grant No. 2019-SFA12, 2022ZY024, 2021-SF-A7-1, Y.D.), and the State Key Laboratory of Earth Surface Processes and Resource Ecology (Grant No. 2022-TS-07, Y.D.).

## Author contributions

Y.D. and H.D. designed the study and planned the analysis. H.D. prepared the basic data, did the data analysis, and wrote the original draft. Q.J. and Y.D. reviewed and edited the manuscript. All authors provided revisions to the manuscript, and approved the final manuscript.

## Competing interests

The authors declare no competing interests.
