## [Peer Review File · Nature Communications]

Assessing coupling interactions in a safe and just operating space for regional sustainabilityREVIEWER COMMENTS

Reviewer #1 (Remarks to the Author):

Assessing coupling interactions in a safe and just operating space for sustainability

Main contributions

The paper analyses China's historical environmental and societal performance (at the regional scale) against the quantified thresholds of a safe and just operating space. The paper presents its assessment, both static (time snapshot) and dynamic (over time). The paper's findings give an overview of the spatial and temporal scale of socioeconomic vs. environmental developments across China and highlight where the trade-offs and synergies exist and where further coordinated strategies would be required.

General feedback

Overall, the connection of planetary boundaries and socio-economic factors (from the SDGs) is a significant topic to study. I can't agree more with respect to need for this research. Given this topic's importance and the interesting case study (China), I expect that the paper will attract broad attention and will become a highly cited article.

I also think that the authors have done a great job in designing a robust methodology, performing a thorough analysis over time and space, and nicely presenting their results (lots of nice figures!). From my point of view, the paper in its current form (with some changes) has a high quality. Having said that, I can see and am willing to suggest some potential ways that the paper can be further improved and reach higher impacts in the field with some more changes and additional work.

Major comments

Clarity of the conceptual framework: In my first reading of the paper, I didn't really get what the conceptual framework is and also didn't understand the meaning and relationships between several terms (e.g., coupling, developing, matrix, etc.). However, some (if not all) of these made sense once I read the results section and looked at methods. I understand the challenge of staying with the word limit and also having a methods section at the end, but I really think the conceptual framework section if rewritten in a clearer way with enough background information, would help reader a lot in better understanding your results. Something to consider.

Research limitations: The authors can acknowledge some of the limitations that the current research has as no paper is complete (some of these limitations might be even raised by others as things that could have been addressed in the current paper). One of these limitations in my view is with respect of the type and number of indicators considered. For example, there are currently one specific indicator under each PB (and therefore the PB results are sensitive to the choice of indicator), but results could change if more or a different indicator(s) were used. Another limitation is with respect to uncertainty in PB and socioeconomic thresholds used. Currently, there is a deterministic threshold and therefore the presented comparison of China's performance against them have also missed the uncertainty in boundary settings. I think the uncertainty in these thresholds becomes even more crucial when we downscale them to the national and regional scale and per capita. Again, the results can vary if the thresholds change. A future research (started by some initial discussions in the current paper) can address the sensitivity of assessments to the diversity of indicators as well as to uncertainty in thresholds.

The local aspect of sustainable development: The progress measurement is highly dependent on knowing and managing a contextualised definition and assessment of sustainable development at sub-national regions, as correctly framed and analysed by the authors. However, there are also important roles to play for the 'communities' as a key actor (sometimes even more important than governments or corporates) and the 'local' scale as a critical level of analysis (as important as - if not higher than - national or regional scale in sustainable development). Highly nuanced local socioeconomic conditions and capacities and the diversity of stakeholder aspirations and interests across local communities, small-scale businesses, and cities can challenge those top-down national approaches to sustainability management (see this perspective article in *One Earth* (Moallemi et al., 2020)). This gives a crucial relevance to the local scale in the study of sustainable development as a context where both impacts and initiators are on the ground. The authors have mentioned the significance of studying at a regional scale to understand the heterogeneities, but I

think they didn't serve justice in discussing the local scale (e.g., local communities, small-scale businesses, cities) and its implications. This doesn't need change in the results or a new analysis. I believe this could be done by highlighting the issue in one or two additional paragraphs in the discussion section.

Detailed comments

L70: 'As' at the beginning is redundant. 'are' to 'is'. Also two important aspects currently missing from this sentence should be mentioned. One is the 'communities' as a key actor (sometimes even more important than governments or corporates) and the other is the local scale as a critical level of analysis (again more important than national or regional in a sustainability context like SDGs). See my last major comment.

L77: 'needs' to 'need'

L83: Instead of saying "these broader conceptual objectives", which may not be clearly known what 'these' are referring to, the authors should specifically mention what their objectives are. I can see these objectives have been mentioned, but in the next paragraph.

L110: there are an excessive number of acronyms which have made the paper difficult to read: SJOS, PB, F-B ESA, CCD, SESs, HWB, etc. Many of them are really unnecessary and should be replaced with their full name for clarity.

L126: the authors refer to "coupling coordination interactions" here and in other parts, what are these 'coupling' and 'coordination', and why can't we just simply say 'interactions'? Maybe I'm not understanding these terms yet as they have not been defined properly so far.

L127: this sentence is not clear and defines development patterns ambiguously. "Development patterns refer to the performance relative to developing and coupling through two-dimensional matrix analysis." I think the reason is that developing and coupling scores haven't been defined yet. The same with the other sentences in this paragraph, e.g., what does "Coupling is categorized according to CCDs" or "While developing is classified based on the trends of CCDs" mean? It should also become clear what low or high degree means in each score. The definition of the four quadrants is not clear and does not say much about what they actually mean in the real world (e.g., what does an uncoupled, underdeveloped pattern look like? does developed/underdeveloped represent both human and environmental developments?). I think this whole paragraph needs to be rewritten with clearer language and a better structure, so to make the understanding of the results easier.

L149: the authors refer to climate change and land-system change and then they discuss the results with reference to these two PBs. But each of these (e.g. climate change) is a very broad area (as opposed to e.g. nitrogen cycle which is more specific) and can include many things (e.g., temperature, CO₂, radiative forcing, etc.) as indicator. I think the authors need to mention the specific indicators they used for four PBs in parentheses here in the main text even though they describe them later in methods.

L190: according to the figure caption, "blue color representing provinces with negative changes". It's not clear if a negative change means decrease in footprints or the deterioration of the footprint (i.e., increase). If it's the former, then why the main text is saying "CO₂ emissions of all provinces have significantly increased, except for Beijing"? Fig 5a shows the opposite, i.e., blue colour for every other region except Beijing.

Fig 5: Is negative change equivalent to deterioration and positive change to improvement? Currently, there could be a confusion that whether negative and positive mean 'decrease and increase in footprints and well-being' or 'deterioration and improvement in footprint and well-being status'. I think the colour map (blue and red) should represent improvement and deterioration (as red often indicates danger, unsafe, undesirable). Another comment here is that why the y-axis here is not the same as y-axis in Fig 4? My understanding is that they are both referring to a same set of indicators.

Fig 3: The figure's blue and green wedges are not drawn accurately. For example, the main text says EN in the world has not reached its threshold, whereas it's reached in China. However, I can see a gap between the EN wedge and the threshold line in both the world's and China's subplots.

Fig S1: there should be additional information in the figure caption, explaining what the circle size represents.

L252: "Dynamic changes in interactions are inconsistent with static results" is unclear. What does a 'dynamic change in interactions' mean? What are 'static' results? They become clearer after I read the whole paragraph and saw the figure, but this sentence which comes at first with no explanatory information can confuse reader.

Figs 6 and 7: Great looking figures, but the units should be mentioned for all axes. The definition of Environmental performance should also be in caption (even though it comes later under methods)

L254: The text says "Trade-offs appear in 4 environmental indicators, except for climate change", this can be confusing a bit as it can mean both trade-offs 'in favour' or 'against' these environmental indicators. If a reader interprets this as 'trade-offs (of any directions) appear in 4 environmental indicators, except for climate change', then this would be seen contradictory to what Fig 7a shows with provinces that traded off climate change with human well-being. I would rewrite this to something like 'socio-economic developments come at the cost of environmental degradation (trade-off) in four out of five indicators...', so I can clearly point to a specific trade-off...

L294, the authors here explain the meaning of different CCD degrees. I think this should have come earlier in this subsection after the first time CCD is mentioned to give reader a better understanding of what the presented numbers imply.

L459: to prioritise.

L456-490: the sentence needs to be rewritten as what it's going to say is unclear.

L618: How were these driving forces selected in the first place for the BRT analysis? I think this needs a bit of background (if they come from other papers) and justification (in terms of their sufficiency in representing various socio-economic-environmental driving forces) as compared to driving forces mentioned in other frameworks (e.g., in SSPs (O'Neill et al., 2017)).

References

Moallemi, E. A., Malekpour, S., Hadjikakou, M., Raven, R., Szetey, K., Ningrum, D., . . . Bryan, B. A. (2020). Achieving the Sustainable Development Goals requires transdisciplinary innovation at the local scale. *One Earth*, 3, 300-313. doi: 10.1016/j.oneear.2020.08.006

O'Neill, B. C., Kriegler, E., Ebi, K. L., Kemp-Benedict, E., Riahi, K., Rothman, D. S., . . . Solecki, W. (2017). The roads ahead: Narratives for shared socioeconomic pathways describing world futures in the 21st century. *Global Environmental Change*, 42, 169-180. doi: <https://doi.org/10.1016/j.gloenvcha.2015.01.004>

I enjoyed reading this paper a lot. I hope you find these comments helpful.

Regards,
Enayat A. Moallemi

Reviewer #2 (Remarks to the Author):

This manuscript provides an assessment of whether China is achieving sustainable development by analysing the progress made towards social well-being and the potential transgression of downscaled planetary boundaries. To this end, a series of numerical and statistical tools are used. The topic is relevant and research efforts aiming at reconciling societal progress with the ecological capacity of the Planet are more than welcome. Despite this, the manuscript suffers from some flaws that this review will cover in a constructive manner in the following paragraphs.

First of all, and acknowledging there is a Methods section, it is recommended that the authors briefly define the tools they use in the relevant sections of the manuscript (e.g., before discussing the results). Note that some of these tools might not be familiar to a general reader, who might also overlook the Methods section. For instance, please, provide a brief explanation of the CCD model and what it does (e.g., at the beginning of section 2.1). This explanation could be similar to the one provided in lines 316-320 for the factors of CCD.

Along the same lines, but even more importantly, the manuscript would benefit from an explanation of the insight the authors are trying to gather by using each of these tools and how their results can be interpreted in the context of the study. For instance, in page 4, please define what coupling and decoupling mean for the present study. It could also help to provide examples for the situations defined by the four Quadrants in Fig. 2. Definitions regarding the physical interpretation of the following concepts would also be welcome: "coordination interactions among

environmental performance and human well-being”, “environmental/social development lag” and “environmental-social synchronization”. In addition, the difference between variables represented in panels a) and b) of Fig. 6 are hard to grasp from the text.

Whilst the methods used seem generally sound, some decisions call for additional justification as otherwise choices made by the authors seem arbitrary. For instance, what is the rationale behind assessing socio-economic and environmental interactions based on the coupling coordination degree (CCD) model? Did anyone else do this in the past? What are these models typically used for?

Please, also explain why only 4 PBs are considered in the assessment when limits have been established for at least 9. In addition, downscaling of PBs based on population is accepted and widely used, yet this is only one of the methods available for downscaling. Acknowledging that the scientific community has not reached an agreement on the best method for such task, this reviewer wonders how sensitive results (and conclusions) are to the downscaling method used. Finally, the literature defines two limits for each PB: one strict value and one that is more loose. The scientific community typically refers to the “uncertainty region” as the one between these two limits, with the size of such region differing from one PB to another. However, the authors define the uncertainty region for the transgression of the PBs as twice the strict limit (regardless of the PB), which seems to neglect the standard practice in the field. Whilst this choice might be acceptable, at least some justification would be expected. Finally, the information provided is very scarce on how footprints were computed, making it difficult to assess whether the indicators used to calculate China’s performance are consistent with those defined in PBs.

Regarding the maximum difference normalization method, this is the first time this reviewer sees it applied in this way: all previous examples contained only the fractional term. What is the literature reference for the particular variant employed here and what is the rationale behind concentrating data around 0.9?

Some interesting and promising results are presented in the manuscript, but they are not discussed in sufficient detail. As an example, there is no attempt in trying to find an explanation for the transgression of the downscaled PBs in some provinces, which hampers the elaboration of specific and useful policies. Similarly, the authors recommend the development of renewable energy as a possible policy choice for regions showing uncoupled and underdeveloped pattern. It could be argued that this statement is vague and seems to neglect the threat posed on the environment by some renewable options (e.g., by bioenergy over land and water use).

The analysis of these temporal results is also very debatable. For instance, Climate Change is presented as an environmental indicator with synergies with Human well-being. As acknowledged by the authors, this is true for 53% of the provinces. However, the analysis of Fig. 7a reveals that all the remaining 47% of the provinces are located in trade-offs Quadrants II and IV, making the conclusion slightly spurious to say the least (i.e., 53% vs 47%). Indeed, results in Fig. 7 are obtained from Sen’s slopes, but it is hard to say whether these are representative or not without information on the corresponding uncertainties. Are the slopes the result of constant trends along the 18 years or do they mask “noisy” data? It is highly recommended that, at the very least, authors plot the temporal series of data in the Supplementary Material (i.e., the information currently provided as Supplementary Tables 4 and 5).

Another question is how do changes in population affect temporal trends shown in Fig. 5 and why is population considered constant between 2050 and 2100 instead of sticking to population prospects as done for the first 30 years? In any case, forecasts beyond 30 years seem unrealistic and unreliable, and the reviewer would question their added value for the manuscript.

Pearson’s correlations in Fig. 6 are very poor (specially the one in panel b), so the authors should be very cautious to draw conclusions from such results.

Finally, some additional minor comments and typos follow:

Page 3, line 70, first sentence is unfinished.

Page 4, line 120, sentence is also unfinished (unless “if” were to be replaced by “whether”).

Page 5, line 170, should be per capita water use.

Fig. 3 caption: "When the green wedge exceeds the dotted line [...]": is not the dotted line the outermost ring? Should not this read as "When the green wedge exceeds the green ring [...]?"

In addition, I would suggest to highlight transgressed PBs and well-being thresholds met in a certain way for quick analysis of the figure (e.g., change colour of label and/or use bold font)

Fig. 4, what does "Unclear" mean and how is this different from "Risky"?

Fig. 5 caption: the reviewer would challenge the use of red for "positive change" across the two subplots and would argue that it would be more intuitive to associate red with "worse performance" (i.e., increase for environmental footprint and decrease for well-being), as in traffic lights.

Page 6, line 221, "indicates" should read "indicated".

Page 6, line 245, "consist" should read "consistent".

Fig. 7, caption: in the last sentence, "out-comes" should read "outcomes".

Page 9, line 353: governments needs "to" pay more.

Page 11, line 479: "pace" should read "space".

Supplementary Fig. 2: consider using black font for labels on yellow and light grey bars.

Reviewer #3 (Remarks to the Author):

In this manuscript the authors quantified a 'safe and just operating space' (SJOS) for China and evaluated how environmental performance and human well-being indicators interact in a SJOS in different regions in China.

I regret not being able to recommend this manuscript for publication in Nature Communication. My main reasons are that the topic and research approach are not novel enough for this journal, and the overall quality of the manuscript is not high enough. As the authors also write, many previous planetary boundaries and social foundation "downscaling" studies have already been published, focusing on different countries and regions. The current manuscript adopts a widely-used 'equal per capita' downscaling approach for China. Nevertheless, in my view, the main contribution of the manuscript is an analysis of regional development patterns, which is interesting as it shows that different parts of China have developed differently regarding environmental and social performance and their interactions. I find the description of methods for this analysis (coupling coordination degree model) difficult to follow but I encourage the authors to highlight and further develop this aspect of the work and submit their manuscript to another journal.

Dear Reviewers,

We much appreciate your constructive and insightful comments and suggestions. We have now carefully and substantively revised the manuscript based on these suggestions. Please see our revision in the manuscript where all changes were highlighted in red. We also provided our point-by-point responses to the comments below.

REVIEWER COMMENTS

Response to Reviewer #1:

Assessing coupling interactions in a safe and just operating space for sustainability

Main contributions

The paper analyses China's historical environmental and societal performance (at the regional scale) against the quantified thresholds of a safe and just operating space. The paper presents its assessment, both static (time snapshot) and dynamic (over time). The paper's findings give an overview of the spatial and temporal scale of socioeconomic vs. environmental developments across China and highlight where the trade-offs and synergies exist and where further coordinated strategies would be required.

Response: Thanks for your positive feedback.

General feedback

Overall, the connection of planetary boundaries and socio-economic factors (from the SDGs) is a significant topic to study. I can't agree more with respect to need for this research. Given this topic's importance and the interesting case study (China), I expect that the paper will attract broad attention and will become a highly cited article. I also think that the authors have done a great job in designing a robust methodology, performing a thorough analysis over time and space, and nicely presenting their results (lots of nice figures!). From my point of view, the paper in its current form (with some changes) has a high quality. Having said that, I can see and am willing to suggest some potential ways that the paper can be further improved and reach higher impacts in the field with some more changes and additional work.

Major comments

1. Clarity of the conceptual framework: In my first reading of the paper, I didn't really get what the conceptual framework is and also didn't understand the meaning and relationships between several terms (e.g., coupling, developing, matrix, etc.). However, some (if not all) of these made sense once I read the results section and looked at methods. I understand the challenge of staying with the word limit and also having a methods section at the end, but I really think the conceptual framework

section if rewritten in a clearer way with enough background information, would help reader a lot in better understanding your results. Something to consider.

Response: Thanks for your great suggestions. This is really a good point on laying out the conceptual framework explicitly upfront, given that the Method section occurred later in the manuscript. We have revised the conceptual figure (Fig.1) and substantially rewritten the conceptual framework section (please see L127-167). We also added additional necessary background information of our framework in Lines 128-135. Moreover, we also provided the definitions and origination of key terminology being used in the manuscript in the text where they were first introduced (e.g., coupling in Line 99; coordination in Line 101, Coupling coordination in Line 103) and summarized these key definitions in Supplementary Table 17. In addition, to help readers to better understand our results and analytical framework, we have added a method flowchart in Supplementary Fig. 1.

2. Research limitations: The authors can acknowledge some of the limitations that the current research has as no paper is complete (some of these limitations might be even raised by others as things that could have been addressed in the current paper). One of these limitations in my view is with respect of the type and number of indicators considered. For example, there are currently one specific indicator under each PB (and therefore the PB results are sensitive to the choice of indicator), but results could change if more or a different indicator(s) were used. Another limitation is with respect to uncertainty in PB and socioeconomic thresholds used. Currently, there is a deterministic threshold and therefore the presented comparison of China's performance against them have also missed the uncertainty in boundary settings. I think the uncertainty in these thresholds becomes even more crucial when we downscale them to the national and regional scale and per capita. Again, the results can vary if the thresholds change. A future research (started by some initial discussions in the current paper) can address the sensitivity of assessments to the diversity of indicators as well as to uncertainty in thresholds.

Response: Great points. We have now added a new section in the manuscript that explicitly stated the limitations of our current research and discussed the potential avenues future research in the Part of "Research limitations" in Lines 571-598.

3. The local aspect of sustainable development: The progress measurement is highly dependent on knowing and managing a contextualised definition and assessment of sustainable development at sub-national regions, as correctly framed and analysed by the authors. However, there are also important roles to play for the 'communities' as a key actor (sometimes even more important than governments or corporates) and the 'local' scale as a critical level of analysis (as important as - if not higher than - national or regional scale in sustainable development). Highly nuanced local socioeconomic conditions and capacities and the diversity of stakeholder aspirations and interests across local communities, small-scale businesses, and cities can

challenge those top-down national approaches to sustainability management (see this perspective article in One Earth (Moallemi et al., 2020)). This gives a crucial relevance to the local scale in the study of sustainable development as a context where both impacts and initiators are on the ground. The authors have mentioned the significance of studying at a regional scale to understand the heterogeneities, but I think they didn't serve justice in discussing the local scale (e.g., local communities, small-scale businesses, cities) and its implications. This doesn't need change in the results or a new analysis. I believe this could be done by highlighting the issue in one or two additional paragraphs in the discussion section.

Response: Thanks for this suggestion on the importance of local communities, stakeholders and contextualized social-economic factors! We fully agree that local thresholds and processes operating could be different from top-down process. The jointly framing of context-specific goals via genuine stakeholder engagement from the bottom-up can complement our framework's guidance for policy-making. We have explored several possible approaches in the Part of "4.3 Opportunities for localizing sustainability" in Lines 551-570 and cited this article in Line 559.

Detailed comments

4.L70: 'As' at the beginning is redundant. 'are' to 'is'. Also two important aspects currently missing from this sentence should be mentioned. One is the 'communities' as a key actor (sometimes even more important than governments or corporates) and the other is the local scale as a critical level of analysis (again more important than national or regional in a sustainability context like SDGs). See my last major comment.

Response: Many thanks! We have revised it and supplemented two missing aspects in this sentence in Lines 69-71.

5.L77: 'needs' to 'need'

Response: Many thanks! We have revised it in Line 76.

6.L83: Instead of saying "these broader conceptual objectives", which may not be clearly known what 'these' are referring to, the authors should specifically mention what then objectives are. I can see these objectives have been mentioned, but in the next paragraph.

Response: Many thanks! These objects refer to the research gaps mentioned in the last paragraph. We have clarified this sentence in Line 83.

7. L110: there are an excessive number of acronyms which have made the paper difficult to read: SJOS, PB, F-B ESA, CCD, SESs, HWB, etc. Many of them are really unnecessary and should be replaced with their full name for clarity.

Response: Many thanks! We have replaced all abbreviations in the text with full names, except for SJOS, CCD, and PB, which have been retained.

8. L126: the authors refer to “coupling coordination interactions” here and in other parts, what are these ‘coupling’ and ‘coordination’, and why can’t we just simply say ‘interactions’? Maybe I’m not understanding these terms yet as they have not been defined properly so far.

Response: Many thanks! We have added the concepts of ‘coupling’ and ‘coordination’ to further explain the meaning of ‘coupling coordination degree’ in Line 99-105. Besides, we have also supplemented some common applications of coupling coordination degree in Lines 105-117.

9. L127: this sentence is not clear and defines development patterns ambiguously. “Development patterns refer to the performance relative to developing and coupling through two-dimensional matrix analysis.” I think the reason is that developing and coupling scores haven’t been defined yet. The same with the other sentences in this paragraph, e.g., what does “Coupling is categorized according to CCDs” or “While developing is classified based on the trends of CCDs” mean? It should also become clear what low or high degree means in each score. The definition of the four quadrants is not clear and does not say much about what they actually mean in the real world (e.g., what does an uncoupled, underdeveloped pattern look like? does developed/underdeveloped represent both human and environmental developments?). I think this whole paragraph needs to be rewritten with clearer language and a better structure, so to make the understanding of the results easier.

Response: Many thanks! We have rewritten this paragraph about development patterns in Lines 143-167.

10. L149: the authors refer to climate change and land-system change and then they discuss the results with reference to these two PBs. But each of these (e.g. climate change) is a very broad area (as opposed to e.g. nitrogen cycle which is more specific) and can include many things (e.g., temperature, CO₂, radiative forcing, etc.) as indicator. I think the authors need to mention the specific indicators they used for four PBs in parentheses here in the main text even though they describe them later in methods.

Response: We have now added the specific control variables of four planetary boundaries in section 1.1 in Lines 184-186.

11. L190: according to the figure caption, “blue color representing provinces with negative changes”. It’s not clear if a negative change means decrease in footprints or the deterioration of the footprint (i.e., increase). If it’s the former, then why the main text is saying “CO₂ emissions of all provinces have significantly increased, except for Beijing”? Fig 5a shows the opposite, i.e., blue colour for every other region except Beijing.

Response: For environmental performance, negative changes mean the deterioration of the footprints (the latter). We have clarified it in the figure caption of Fig. 5 in Lines 1022-1024.

12. Fig 5: Is negative change equivalent to deterioration and positive change to improvement? Currently, there could be a confusion that whether negative and positive mean ‘decrease and increase in footprints and well-being’ or ‘deterioration and improvement in footprint and well-being status’. I think the colour map (blue and red) should represent improvement and deterioration (as red often indicates danger, unsafe, undesirable). Another comment here is that why the y-axis here is not the same as y-axis in Fig 4? My understanding is that they are both referring to a same set of indicators.

Response: Good point. We have revised the colors of Fig. 5 and added additional explanation in the figure caption. In the color map, blue represents improvement, and red represents deterioration. We have clarified the meanings of color in the capture of Fig. 5. Besides, we have revised the y-axis of Fig. 5 to be consistent with the y-axis in Fig. 4.

13. Fig 3: The figure’s blue and green wedges are not drawn accurately. For example, the main text says EN in the world has not reached its threshold, whereas it’s reached in China. However, I can see a gap between the EN wedge and the threshold line in both the World’s and China’s subplots.

Response: Many thanks! We have modified the Fig. 3 to make it more accurate, and highlighted the colors of the wedges of transgressed planetary boundaries and social thresholds met for quick analysis of the figure.

14. Fig S1: there should be additional information in the figure caption, explaining what the circle size represents.

Response: We have explained what the circle size represents in the caption of Fig. S5.

15. L252: “Dynamic changes in interactions are inconsistent with static results” is unclear. What does a ‘dynamic change in interactions’ mean? What are ‘static’ results? They become clearer after I read the whole paragraph and saw the figure, but this sentence which comes at first with no explanatory information can confuse reader.

Response: We have explained the meaning of ‘dynamic results’ and ‘static results’ in the first paragraph of the Part 1.3 in Lines 275-279.

16. Figs 6 and 7: Great looking figures, but the units should be mentioned for all axes. The definition of Environmental performance should also be in caption (even though it comes later under methods)

Response: Many thanks! We have added the units for all axes in Fig. 6. As for Fig. 7, environmental performance (x-axis) and human well-being (y-axis) are both dimensionless ratios. We have added the definitions of environmental performance and human well-being in the captions of Fig. 6 and Fig. 7.

17. L254: The text says “Trade-offs appear in 4 environmental indicators, except for climate change”, this can be confusing a bit as it can mean both trade-offs ‘in favour’ or ‘against’ these environmental indicators. If a reader interprets this as ‘trade-offs (of

any directions) appear in 4 environmental indicators, except for climate change’, then this would be seen contradictory to what Fig 7a shows with provinces that traded off climate change with human well-being. I would rewrite this to something like ‘socio-economic developments come at the cost of environmental degradation (trade-off) in four out of five indicators....’, so I can clearly point to a specific trade-off...

Response: We have revised the text in Lines 295-297.

18. L294, the authors here explain the meaning of different CCD degrees. I think this should have come earlier in this subsection after the first time CCD is mentioned to give reader a better understanding of what the presented numbers imply.

Response: Many thanks! We have explained the meaning of the value of coupling coordination degree in the first paragraph of the Part 2.1, Lines 316-323.

19. L459: to prioritise.

Response: We have revised it in Line 542.

20. L456-490: the sentence needs to be rewritten as what it’s going to say is unclear.

Response: We have rewritten this sentence in Lines 620-623.

21. L618: How were these driving forces selected in the first place for the BRT analysis? I think this needs a bit of background (if they come from other papers) and justification (in terms of their sufficiency in representing various socio-economic-environmental driving forces) as compared to driving forces mentioned in other frameworks (e.g., in SSPs (O’Neill et al., 2017)).

Response: Many thanks! We have added the reasons for selecting these driving factors and compared them with the drivers mentioned in O’Neill et al. (2017), in Lines 762-767.

References

Moallemi, E. A., Malekpour, S., Hadjidakou, M., Raven, R., Szetey, K., Ningrum, D., . . . Bryan, B. A. (2020). Achieving the Sustainable Development Goals requires transdisciplinary innovation at the local scale. *One Earth*, 3, 300-313. doi: 10.1016/j.oneear.2020.08.006

O’Neill, B. C., Kriegler, E., Ebi, K. L., Kemp-Benedict, E., Riahi, K., Rothman, D. S., . . . Solecki, W. (2017). The roads ahead: Narratives for shared socioeconomic pathways describing world futures in the 21st century. *Global Environmental Change*, 42, 169-180. doi: <https://doi.org/10.1016/j.gloenvcha.2015.01.004>

I enjoyed reading this paper a lot. I hope you find these comments helpful.

Regards,

Enayat A. Moallemi

Response to Reviewer #2:

This manuscript provides an assessment of whether China is achieving sustainable development by analysing the progress made towards social well-being and the potential transgression of downscaled planetary boundaries. To this end, a series of numerical and statistical tools are used. The topic is relevant and research efforts aiming at reconciling societal progress with the ecological capacity of the Planet are more than welcome. Despite this, the manuscript suffers from some flaws that this review will cover in a constructive manner in the following paragraphs.

Response: Thanks for all your constructive comments.

1. First of all, and acknowledging there is a Methods section, it is recommended that the authors briefly define the tools they use in the relevant sections of the manuscript (e.g., before discussing the results). Note that some of these tools might not be familiar to a general reader, who might also overlook the Methods section. For instance, please, provide a brief explanation of the CCD model and what it does (e.g., at the beginning of section 2.1). This explanation could be similar to the one provided in lines 316-320 for the factors of CCD.

Response: This is a good suggestion that will help the flow and better illustrate the results given that the Method section occurred at the end of the manuscript! We now have woven in the manuscript where appropriate that briefly defined the tools and approaches used early in the manuscript before going to the details of the results. For example, we have added explanation of the CCD model in section 2.1 in Lines 309-314, and Mann-kendall trend test and Sen's slope in section 2.2 in Lines 354-355.

2. Along the same lines, but even more importantly, the manuscript would benefit from an explanation of the insight the authors are trying to gather by using each of these tools and how their results can be interpreted in the context of the study. Definitions regarding the physical interpretation of the following concepts would also be welcome: “coordination interactions among environmental performance and human well-being”, “environmental/social development lag” and “environmental-social synchronization”.

Response: In this revision, we have explained each tool used in method section and added the method flowchart Fig. S1 that helped explain our overall analytical framework. For example, Mann-kendall trend test in Lines 684-687, coupling coordination degree model in Lines 719-720, boosted regression tree method in Lines 757-759. Also per suggestions from reviewer #1, we also provided detailed definitions of key terminologies used in the manuscript upfront to avoid any confusions and summarized these key definitions in Supplementary Table 17.

3. For instance, in page 4, please define what coupling and decoupling mean for the present study. It could also help to provide examples for the situations defined by the four Quadrants in Fig. 2.

Response: We have explained the meaning of ‘coupling’ and ‘decoupling’ in Lines 150-152, and rewritten the development patterns of conceptual framework section in Lines 146-167. And we have added the definitions of ‘coupling’ and ‘coordination’ in Lines 99-102.

4. Definitions regarding the physical interpretation of the following concepts would also be welcome: “coordination interactions among environmental performance and human well-being”, “environmental/social development lag” and “environmental-social synchronization”.

Response: Many thanks! We have added the meaning of the value of coupling coordination degree in Lines 316-318, and explained how to distinguish ‘environmental development lag’, ‘social development lag’ and ‘environmental-social synchronization’ types in Lines 318-323.

5. In addition, the difference between variables represented in panels a) and b) of Fig. 6 are hard to grasp from the text.

Response: Variables in Fig. 6 panel a represent the numbers of social thresholds reached and numbers of environmental boundaries operated within in each province. Variables in Fig. 6 panel b represent human well-being and environmental performance in each province. We have added these definitions of ‘environmental performance’ and ‘human well-being’ in the caption of Fig. 6 to clarify.

6. Whilst the methods used seem generally sound, some decisions call for additional justification as otherwise choices made by the authors seem arbitrary. For instance, what is the rationale behind assessing socio-economic and environmental interactions based on the coupling coordination degree (CCD) model? Did anyone else do this in the past? What are these models typically used for?

Response: Thanks for your comments! We agree that it is important to provide further justification on the coupling coordination degree (CCD) model used for our analyses, as well as some background from the literature. In the revised manuscript, we have now added the conceptual foundation of coupling and coordination (see Lines 99-117 in the section of Introduction), its definition and application in previous empirical studies in the general field of sustainability science, as well as how to interpret the results from the CCD model. In addition, we also have added the advantages and the reasons for choosing each tool in method section and added a method flowchart (Fig. S1) to demonstrate the overall data flow and analytical processes.

7. Please, also explain why only 4 PBs are considered in the assessment when limits have been established for at least 9.

Response: We have added the justification for choosing these 4 planetary boundaries in Lines 27-43 in the Supplementary Information.

8. In addition, downscaling of PBs based on population is accepted and widely used, yet this is only one of the methods available for downscaling. Acknowledging that the scientific community has not reached an agreement on the best method for such task, this reviewer wonders how sensitive results (and conclusions) are to the downscaling method used.

Response: This is a great point as also raised by reviewer #1. In our revised manuscript, we have explained the reasons for allocating the PBs based on per capita shares in Lines 47-53 in Supplementary Information. Besides, to examine the robustness of our results to the downscaling method used, we further performed a sensitivity analysis in Supplementary Information, in lines 246-283. Parameter sensitivity analysis refers to the analysis and measurement of their degree of influence on the results and the degree of sensitivity. Realizing that downscaling methods and allocation schemes are still an active area of research, we also acknowledged in the manuscript (see Lines 581-588) with a new section (i.e., 4.4) on research limitation, where we stated how downscaling approach might play a role in affecting the results.

9. Finally, the literature defines two limits for each PB: one strict value and one that is more loose. The scientific community typically refers to the “uncertainty region” as the one between these two limits, with the size of such region differing from one PB to another. However, the authors define the uncertainty region for the transgression of the PBs as twice the strict limit (regardless of the PB), which seems to neglect the standard practice in the field. Whilst this choice might be acceptable, at least some justification would be expected.

Response: Many thanks! We have revised the ‘uncertainty state’ to ‘potentially unsafe state’ in the main text and Fig. 4 and added the details of the four categories defined with environmental performance in Table S3 in Supplementary Information.

10. Finally, the information provided is very scarce on how footprints were computed, making it difficult to assess whether the indicators used to calculate China’s performance are consistent with those defined in PBs.

Response: To address this comment, we have supplemented additional information on how to calculate the footprints and downscaling PBs in Section 1.1-1.4 in Supplementary information. Besides, we also added additional details of social indicators and social thresholds in Section 2 in Supplementary Information.

11. Regarding the maximum difference normalization method, this is the first time this reviewer sees it applied in this way: all previous examples contained only the fractional term. What is the literature reference for the particular variant employed

here and what is the rationale behind concentrating data around 0.9?

Response: Many thanks! We have revised our data normalized method in Lines 713-714 and updated all corresponding results, including main text, tables and figures.

12. Some interesting and promising results are presented in the manuscript, but they are not discussed in sufficient detail. As an example, there is no attempt in trying to find an explanation for the transgression of the downscaled PBs in some provinces, which hampers the elaboration of specific and useful policies. Similarly, the authors recommend the development of renewable energy as a possible policy choice for regions showing uncoupled and underdeveloped pattern. It could be argued that this statement is vague and seems to neglect the threat posed on the environment by some renewable options (e.g., by bioenergy over land and water use).

Response: We appreciate this thoughtful comment and suggestions! We have improved the discussion section, especially the interpretation on the mechanisms of spatial and temporal patterns on environmental performance, human well-being and CCD in section 4.1 (see Lines 422-429, Lines 452-465) and the elaboration of policy recommendations for four development patterns in section 4.2 (see Lines 474-550). For policy recommendations, we have added the characteristics of provinces in each pattern, further explained possible reasons and proposed more reasonable suggestions.

13. The analysis of these temporal results is also very debatable. For instance, Climate Change is presented as an environmental indicator with synergies with Human well-being. As acknowledged by the authors, this is true for 53% of the provinces. However, the analysis of Fig. 7a reveals that all the remaining 47% of the provinces are located in trade-offs Quadrants II and IV, making the conclusion slightly spurious to say the least (i.e., 53% vs 47%). Indeed, results in Fig. 7 are obtained from Sen's slopes, but it is hard to say whether these are representative or not without information on the corresponding uncertainties. Are the slopes the result of constant trends along the 18 years or do they mask "noisy" data? It is highly recommended that, at the very least, authors plot the temporal series of data in the Supplementary Material (i.e., the information currently provided as Supplementary Tables 4 and 5).

Response: Many thanks! We have plotted the temporal series of environmental performance and human well-being in Fig. S3 and S4, provided as the supplementary Information for Table S7 and S8. Also, to clarify the results in Fig. 7, we further distinguished the points on the axes and added the figures for the percentage of provinces in each quadrant in Fig. 7 and Table S9. To address this comment, we further made revisions in the main text in Lines 300-306.

14. Another question is how do changes in population affect temporal trends shown in

Fig. 5 and why is population considered constant between 2050 and 2100 instead of sticking to population prospects as done for the first 30 years? In any case, forecasts beyond 30 years seem unrealistic and unreliable, and the reviewer would question their added value for the manuscript.

Response: Many thanks for this valuable suggestion! We have updated the population between 2050 and 2100 using the interpolated total population data in medium fertility from United Nations Population Division, and revised all corresponding results, including main text, tables (Tables S2, S4-S5, S7, S9-S12) and figure (Figs. 3-10, Figs. S3-S7). We have added the details of downscaling in Lines 627-635 in manuscript and the section 1.1 in Supplementary Information.

15. Pearson' correlations in Fig. 6 are very poor (specially the one in panel b), so the authors should be very cautious to draw conclusions from such results.

Response: Many thanks for this valuable suggestion! We have refitted the relationships between comprehensive development levels of human well-being and environmental performance using linear-logarithmic function in Fig.6b. We have rewritten the results from Fig.6 in Lines 290-292 and added additional explanation in the figure caption of Fig.6.

Finally, some additional minor comments and typos follow:

16. Page 3, line 70, first sentence is unfinished.

Response: Many thanks! We have revised this sentence in Lines 69-71.

17. Page 4, line 120, sentence is also unfinished (unless “if” were to be replaced by “whether”).

Response: We have revised ‘if’ to ‘whether’ in Line 183.

18. Page 5, line 170, should be per capita water use.

Response: We have revised it in Line 204.

19. Fig. 3 caption: “When the green wedge exceeds the dotted line [...]”: is not the dotted line the outermost ring? Should not this read as “When the green wedge exceeds the green ring [...]”?

Response: Many thanks! We have revised it in the caption of Fig. 3, in lines 1004-1006.

20. In addition, I would suggest to highlight transgressed PBs and well-being thresholds met in a certain way for quick analysis of the figure (e.g., change colour of label and/or use bold font)

Response: Good point! We highlighted the colors of the wedges of transgressed planetary boundaries and social thresholds met in Fig. 3 and added additional explanation in the capture of Fig. 3 in Lines 1003-1004.

21. Fig. 4, what does “Unclear” mean and how is this different from “Risky”?

Response: We have revised ‘unclear state’ to ‘potentially unsafe’, ‘risky state’ to ‘clearly unsafe state’, and added the meanings of the four categories defined with environmental performance in Table S3 in Supplementary Information.

22. Fig. 5 caption: the reviewer would challenge the use of red for “positive change” across the two subplots and would argue that it would be more intuitive to associate red with “worse performance” (i.e., increase for environmental footprint and decrease for well-being), as in traffic lights.

Response: Many thanks! We have revised the colors of Fig. 5 and added additional explanation in the figure caption. In the color map, blue represents improvement, and red represents deterioration.

23. Page 6, line 221, “indicates” should read “indicated”.

Response: We have revised it in Line 257.

24. Page 6, line 245, “consist” should read “consistent”.

Response: We have revised it in Line 285.

25. Fig. 7, caption: in the last sentence, “out-comes” should read “outcomes”.

Response: Many thanks! We have revised ‘out-comes’ to ‘outcomes’ in Line 1045.

26. Page 9, line 353: governments needs “to” pay more.

Response: We have revise it in Line 407.

27. Page 11, line 479: “pace” should read “space”.

Response: We have revise it in Line 614.

28. Supplementary Fig. 2: consider using black font for labels on yellow and light grey bars.

Response: Many thanks! We have revised labels to black font in Fig. S6.

Response to Reviewer #3:

In this manuscript the authors quantified a ‘safe and just operating space’ (SJOS) for China and evaluated how environmental performance and human well-being indicators interact in a SJOS in different regions in China.

I regret not being able to recommend this manuscript for publication in Nature Communication. My main reasons are that the topic and research approach are not novel enough for this journal, and the overall quality of the manuscript is not high enough. As the authors also write, many previous planetary boundaries and social foundation “downscaling” studies have already been published, focusing on different

countries and regions. The current manuscript adopts a widely-used ‘equal per capita’ downscaling approach for China. Nevertheless, in my view, the main contribution of the manuscript is an analysis of regional development patterns, which is interesting as it shows that different parts of China have developed differently regarding environmental and social performance and their interactions. I find the description of methods for this analysis (coupling coordination degree model) difficult to follow but I encourage the authors to highlight and further develop this aspect of the work and submit their manuscript to another journal.

***Response:** Thank you for your constructive comments and suggestions to further improve this work! In our revised manuscript, we have made clear on the novel contribution of this research, which is the proposed development pattern on the basis of coupling relationships between environmental performance and human wellbeing and their changes. We fully agree that the concept of planetary boundary (PB) and social foundations is not new, and there has been a number of empirical studies focusing on different systems or regions. However, to the best of our knowledge, few studies have taken a step further to use the social and environmental indicators related to PB and social foundations (in the context of SJOS) to determine coupling coordination degree across spatial and temporal scales, which further informs development patterns and corresponding policy recommendations for achieving sustainability. Such analyses are not always feasible nor have been common or published for other countries due to the need of high resolution of social and environmental datasets. Another contribution of our work is that our analyses also accounted for the complex patterns across spatial scales (e.g., provinces, regions, and national) and temporal dynamics and their interactions, whereas the major of prior studies have focused on either one spatial scale and even fewer has explicitly analyzed interactions between spatial and temporal patterns for both environmental and social variables. Further, this work also has important policy implications, given that (1) China is the second largest economies with important international responsibilities in terms of achieving sustainable development, reducing environmental footprints, addressing social inequality, and combating climate change; (2) how our results on the coupling coordination relationships between environmental performance and social foundations would inform regional and national policies pertaining to sustainable management and development. We anticipate that our framework can also be applied to other countries and regions to achieve more resilient and sustainable development.*

To address another comment on the methods, per suggestions from reviewer #1 and #2, we have made substantive revisions on the clarification the key concepts and terminology used in the manuscript. We also provided additional details to clarify the usage and justification of our methods both earlier in the manuscript (for better understanding the results) and in the Method and Supplementary Information sections.

REVIEWER COMMENTS

Reviewer #1 (Remarks to the Author):

Thank you for your efforts in addressing all the reviewer comments. The paper now reads really well and has several beautiful visualisations! I think the paper will be a significant contribution method-wise and also by providing important insights from China's performance against PBs and socio-economic indicators.

Reviewer #2 (Remarks to the Author):

This report is for the second review round of the manuscript "Assessing coupling interactions in a safe and just operating 1 space for sustainability", which builds a methodological framework aiming at analysing the reconciliation of human well-being with planetary boundaries (PBs) in a safe and just operating space (SJOS).

In comparison to the first version of the manuscript, reviewed in July of this year, there have been immense improvements to the work, most notably with respect to the description of the methods and data used, as well as the inclusion of population prospects in the calculations. This has been a huge amount of work and it has to be appreciated that the quality of the paper has greatly benefitted from these improvements.

There are some open questions with respect to my previous comments, which need some clarification (see below). All else has been addressed to my full satisfaction and I thank the author team for their reply letter in this context. Apart from this, the new inserts have arisen new important questions, which I also pose for authors' consideration.

I will start with my two main concerns at this point. The first one is the lack of a continuous narrative between sections 1.1.1 and section 3 (i.e., covering Figs. 4-8). In this central part of the manuscript, the reader is faced with several analyses, which (1) seem repetitive and (2) are not discussed in a constructive way since all the insightful discussion relating the results with regional realities appears only in section 4. For instance, what is the added value of Section 3 compared to previous discussions in Section 2? It feels like a repetition. With this comment, I am not suggesting that the analyses presented are wrong or not interesting, but many readers will not grasp the subtle differences between some of the facets the authors try to study through Figs. 4-8. In this context, I would suggest the authors carefully revise this section bearing in mind the following considerations: (1) what is the added value of each new figure with respect to the previous (or later) figures? (2) Is it worth having all these figures in the main manuscript or can some be moved to the supplementary material? (3) motivate the need for each new analysis: what is the new insight the authors are looking for and could not be grasped with previous analyses? (4) Consider discussing individual implications of results in each figure in the relevant sections, instead of leaving them all to section 4.

My second main concern can be found in section 1.4 of the Supplementary material. The authors appropriately explain that the PB of the N cycle stands at 62Tg/yr of N from industrial and intended biological fixation. However, they next say "Allocation of nitrogen fertilizer applied to cropland has been selected as the control variable", which suggests that they compare the nitrogen flows from intended biological fixation (i.e., disregarding nitrogen flows from industry) to the whole PB (which also considers flows stemming from the chemical industry). If this is true, calculations for this PB would be wrong. There is a workaround to solve this issue, as explained in "Algunaibet et al., Correction: Powering sustainable development within planetary boundaries, et al., Energy Environ. Sci., 2019, 12, 1890–1900". In essence, it consists of splitting the 62Tg/yr PB into chemical and agricultural contributions based on the current burdens of the two sectors on this PB, and then use as ceiling only the share corresponding to agricultural flows (i.e., intended biological fixation).

Some more-specific comments follow.

One important aspect is related with the novelty of the contribution, which is not clearly declared in the manuscript. The new introduction points towards several works using similar tools to address similar problems, and it could be helpful to explicitly describe what makes this contribution stand out from the rest.

Page 4, lines 152-158. It is now unclear what is the difference between the coupling level (y-axis)

and the development level (x-axis), since the later seems to be defined based on the former (e.g., from lines 154-157, "Developed regions tend to have an increasing level of coupling, whereas underdeveloped regions show a trend towards a more uncoupled direction"). Is development level related with the achievement of social foundations and/or operation within the environmental ceiling defined by the PBs?

I appreciate the addition of Supplementary Fig. 3 to address my original comment on the definition of the uncertain region (i.e., potentially unsafe) of the PBs as 200% their safe threshold. However, the spirit of the comment is still overlooked, that is, the fact that these limits are not the ones used by the main literature on PBs. As an example, even the left-most panel used by the authors in Fig. 1 is not based on their definition (i.e., 200%)! Please, either adopt the common approach whereby the uncertain region of each PB is tailored, or explicitly comment on this difference which otherwise may confuse several readers.

Results on Fig. 6b are not clear (discussion in lines 290-292). The figure is presented as the normalised version of 6a, but if that would be the case, then the figure would look exactly the same but with different values. How are the (normalized) values plotted in Fig. 6b calculated? Is it the average transgression (if any) among the five PBs considered? Please, clarify.

Discussion on Fig. 7 (lines 293-306) is not clear neither. Trade-off relationships are suggested between social well-being and all PBs except for climate change, but "similar" values are obtained by adding the percentage of regions in Quadrants I-III vs in Quadrants II-IV (e.g., 40% vs 50% for freshwater use, or 50% vs 43% for P cycle). In addition, percentages add up to 90% instead of 100% in Fig. 7 (please, check).

Lines 184-188 could potentially contain a reference to the appropriate section of supplementary material. Similarly, section 4.4 on research limitations should include a reference to section 3 in the supplementary material, as well as briefly comment on the findings of this section. In general, all the parts in the supplementary material should be referred to somewhere in the main manuscript.

Conclusions are slightly vague in terms of significance and would benefit from including specific examples on the trends observed and guidelines provided. This way, they will demonstrate how the proposed framework can inform policymakers and help them develop more effective policies. This could also help clarifying the novelty of the manuscript.

While the explanation in lines 21-43 of the Supplementary Material is interesting, it still does not reflect on the rationale behind selecting only four (five) of the nine PBs quantified to date. Please, address this issue.

Results in Supplementary Fig. 2 are only described, but one would expect that their *implications* were also addressed. For instance, what is the impact of the high sensitivity of climate change and the P cycle in Supplementary Fig. 2a? Does this change any of the conclusions in the main manuscript? This would not be a negative thing per se, but would simply deserve discussion. The same applies to results in Supplementary Fig. 2b.

Minor comment and typos:

Line 477, should read [...] is "in" contrast to [...].

Avoid contract forms in formal/technical language (see "doesn't" in lines 684 or "It's" in line 685).

Data availability: Tables S14-S16 should also be referred to in the specific points of the manuscript where these data is relevant (e.g., Table S16 somewhere in lines 760-771).

Supplementary Figs. 3 and 4. Please indicate clearly the variables depicted in the Y-axes and their units.

Suggestions:

Lines 178-181, clarify that two PBs are defined for biogeochemical flows, as otherwise the unfamiliar reader might be confused when reading "four planetary boundaries" and then finding five sectors in Fig. 3. The same applies to lines 636-638 in the Methods section (and other places in the manuscript).

In Fig. 3, I would suggest using consistent colours between social and environmental aspects: if grey is "bad" for social, let it be "bad" for the environment, too (i.e., paint in green only the PBs which are not transgressed, and leave grey for those transgressed).

In lines 331-339, briefly mention that the "V1-9 code" refers to the combination of different coordination degrees with social/environmental lag.

The discussion about the Balanced strategy (starting at line 513) is insightful and interesting.

When talking about energy exporters regions, it might be worth mentioning that a different picture

could emerge by measuring impacts through a consumption-based approach as done in previous works (e.g., Davis and Caldeira, PNAS. 2010, 107, 5687-5692; Pozo et al., JoCP 2020, 270, 121828).

Supplementary Table 13 could be more informative if presented as a matrix.

Reviewer #3 (Remarks to the Author):

This work has potential for a very interesting analysis of regional differences in environmental performance and social development and their driving factors in China. However, when reading the revised version, I don't see that the authors would have improved the manuscript and their analysis significantly enough for me to be able to recommend it as it is for publication in Nature Communications. My main points are the following:

Motivation of the study - The Introduction lacks a clear statement of the research challenge/question. It only lists objectives of the paper (i.e. what is being done). Because of this, it remains unclear for the reader what the purpose of the paper is and what new knowledge this paper aims to produce.

Frameworks - Related to the unclear aim, it is also unclear why the authors have chosen the planetary boundaries framework for their study. Is it really the best framework if you are looking at how regional/provincial development and environmental impacts are connected, in particular if you are taking a territorial/production-based approach for estimating the environmental impacts? The exercise of downscaling the planetary boundaries framework to national/regional level is often helpful when assessing a country's contribution to the global environmental pressures, including trade-related impacts. Why are the authors not including environmental indicators related to air and water pollution and resource use?

Data - It is also unclear if the authors use "territorial/production-based" or "consumption-based" values for the environmental footprints. In many previous planetary boundaries downscaling studies, both production and consumption-based footprints are calculated and compared to give a fuller picture of a country's impact. It's also unclear when the authors are using values from O'Neill et al (2018) paper (see Fig 3 in the manuscript) or values from other sources.

Analysis and results - The authors use several methods, including statistical analysis, coupling coordination model and boosted regression three method based on machine learning. More explanation of why the authors have designed and implemented the study as it is, instead of merely telling what they have done, would have been welcome. It is also hard to understand how the authors have used these methods. The analysis of the produced results could have been done much more in depth.

Discussion - the Discussion section should discuss the results better, their meaning and insights from them. And, in particular, the results from the machine learning method and related discussion about the driver factors of the different development patterns could be further developed much more. The Discussion section also does not adequately situate the findings into the existing literature in development and sustainability research.

Writing and style - the overall writing, structuring and organisation of the text should be improved to make it easier for the reader to follow the argumentation, findings and conclusions.

Dear Reviewers,

We much appreciate your constructive and insightful comments and suggestions. We have now carefully and substantively revised the manuscript based on these suggestions. Please see our detailed revisions in the manuscript where all changes were highlighted in red font. We also provided our point-by-point responses to the comments below.

REVIEWER COMMENTS

Reviewer #1 (Remarks to the Author):

Thank you for your efforts in addressing all the reviewer comments. The paper now reads really well and has several beautiful visualisations! I think the paper will be a significant contribution method-wise and also by providing important insights from China's performance against PBs and socio-economic indicators.

Response: Thanks for your positive feedback.

Reviewer #2 (Remarks to the Author):

This report is for the second review round of the manuscript “Assessing coupling interactions in a safe and just operating 1 space for sustainability”, which builds a methodological framework aiming at analysing the reconciliation of human well-being with planetary boundaries (PBs) in a safe and just operating space (SJOS). In comparison to the first version of the manuscript, reviewed in July of this year, there have been immense improvements to the work, most notably with respect to the description of the methods and data used, as well as the inclusion of population prospects in the calculations. This has been a huge amount of work and it has to be appreciated that the quality of the paper has greatly benefitted from these improvements.

Response: Thank you for your positive feedback related to our previous revision.

There are some open questions with respect to my previous comments, which need some clarification (see below). All else has been addressed to my full satisfaction and I thank the author team for their reply letter in this context. Apart from this, the new inserts have arisen new important questions, which I also pose for authors' consideration.

Response: Thanks for all your constructive and insightful comments. Please see our detailed response and revisions below.

I will start with my two main concerns at this point.

1. The first one is the lack of a continuous narrative between sections 1.1.1 and section 3 (i.e., covering Figs. 4-8). In this central part of the manuscript, the reader is faced with several analyses, which (1) seem repetitive and (2) are not discussed in a

constructive way since all the insightful discussion relating the results with regional realities appears only in section 4. For instance, what is the added value of Section 3 compared to previous discussions in Section 2? It feels like a repetition. With this comment, I am not suggesting that the analyses presented are wrong or not interesting, but many readers will not grasp the subtle differences between some of the facets the authors try to study through Figs. 4-8. In this context, I would suggest the authors carefully revise this section bearing in mind the following considerations: (1) what is the added value of each new figure with respect to the previous (or later) figures? (2) Is it worth having all these figures in the main manuscript or can some be moved to the supplementary material? (3) motivate the need for each new analysis: what is the new insight the authors are looking for and could not be grasped with previous analyses? (4) Consider discussing individual implications of results in each figure in the relevant sections, instead of leaving them all to section 4.

***Response:** Thanks for your valuable comments and thoughtful questions for us to ponder, which are great for us to think through ways to distill most important results of this work as well as streamline the flow and structure to make it more accessible to the audience. Based on these suggestions and considerations, we have substantially reorganized the structure of the Results section to reduce repetition as commented. In addition, we also modified the Supplementary Information section, and moved certain materials (e.g., Fig. 6 and Fig. 7) to the Supplementary Information so as to only retain the key results in the Main Text. We also consolidated several sections (e.g., Section 2 and 3 in the previous revision) for minimizing any redundancies and improving clarity. Further, to provide a continuous narrative, we have added a statement of purpose for each section before going to the details of the results and summarized the discussion related to the results of each section. Finally, per your suggestion, in the revised manuscript, we also discussed the implications of results of figures in each of the relevant sections.*

2. My second main concern can be found in section 1.4 of the Supplementary material. The authors appropriately explain that the PB of the N cycle stands at 62Tg/yr of N from industrial and intended biological fixation. However, they next say “Allocation of nitrogen fertilizer applied to cropland has been selected as the control variable”, which suggests that they compare the nitrogen flows from intended biological fixation (i.e., disregarding nitrogen flows from industry) to the whole PB (which also considers flows stemming from the chemical industry). If this is true, calculations for this PB would be wrong. There is a workaround to solve this issue, as explained in “Algunaibet et al., Correction: Powering sustainable development within planetary boundaries, et al., Energy Environ. Sci., 2019, 12, 1890–1900”. In essence, it consists of splitting the 62Tg/yr PB into chemical and agricultural contributions based on the current burdens of the two sectors on this PB, and then use as ceiling only the share corresponding to agricultural flows (i.e., intended biological fixation).

***Response:** Thanks for your great suggestions. We have revised the downscaled*

method for nitrogen cycle in Method section of manuscript (please see Lines 737-742), added more details in Supplementary information (please see Lines 168-174), and updated all corresponding results, including main text, tables and figures (please see Fig. 3-8, Fig. S12-S21).

Some more-specific comments follow.

3. One important aspect is related with the novelty of the contribution, which is not clearly declared in the manuscript. The new introduction points towards several works using similar tools to address similar problems, and it could be helpful to explicitly describe what makes this contribution stand out from the rest.

***Response:** Great suggestions. In our revised manuscript, we emphasized that global sustainability will inevitably be actualized through firstly achieving regional sustainability. Our work aims to propose a quantitative and operationalizable framework, on the basis of SJOS and SDGs and from the conception of the strong sustainability paradigm, to assess regional sustainability across scales for effective sustainable management. While we focused China as a demonstration case, our analytical framework is broadly applicable to other regions or nations to support policy-making and implementation of targeted strategies to promote stronger sustainability. The novel aspects of this research include: 1) explicitly addressing spatial and temporal variations and across-scale dynamics in assessing sustainability relative to SJOS; 2) introducing an innovative coupling coordination degree analysis based on the sustainability assessment to conceptualize and account for interactions between environmental performance and human well-being goals; 3) taking the SDGs as justifiable target thresholds for social indicators in assessing sustainability on the basis of SJOS. To address these suggestions, we have further elaborated the research questions (please see Lines 87-91), explicitly clarified the novel contributions of our study compared to previous research, and substantially revised the Introduction section to highlight how our work differs from previous research and contributes to the field of sustainability science (please see Lines 101-116). In addition, we have also summarized the contributions of this work in the Conclusion section (please see Lines 681-688).*

4. Page 4, lines 152-158. It is now unclear what is the difference between the coupling level (y-axis) and the development level (x-axis), since the later seems to be defined based on the former (e.g., from lines 154-157, “Developed regions tend to have an increasing level of coupling, whereas underdeveloped regions show a trend towards a more uncoupled direction”). Is development level related with the achievement of social foundations and/or operation within the environmental ceiling defined by the PBs?

***Response:** The coupling level is quantified by the magnitude of CCD, whereas the development level is quantified by changing trends of CCD (i.e., direction of CCD). That is, the development level represents the direction of changes in the coupling*

level. In our study, higher levels of development indicate becoming more synergies (i.e., coupling) over time whereas low values mean becoming more trade-offs (i.e., decoupling) between achieving environmental performance and human well-being goals. To clarify this confusion, we have added more explanations in the manuscript (please see Lines 176-178) and in the caption of Fig.2 (please see Lines 1114-1116).

5. I appreciate the addition of Supplementary Fig. 3 to address my original comment on the definition of the uncertain region (i.e., potentially unsafe) of the PBs as 200% their safe threshold. However, the spirit of the comment is still overlooked, that is, the fact that these limits are not the ones used by the main literature on PBs. As an example, even the left-most panel used by the authors in Fig. 1 is not based on their definition (i.e., 200%)! Please, either adopt the common approach whereby the uncertain region of each PB is tailored, or explicitly comment on this difference which otherwise may confuse several readers.

Response: *Thanks for your constructive comment and further elaborating on your previous point! To address this comment, we have added explanation of uncertainty regions in Supplementary Information (please see Lines 214-223), revised the category definition of environmental performance in Table. S3, and tailored the uncertain region of each PB in Table. S4. In addition, we have added the comparison of consumptive and territorial environmental performance based on the uncertain zones of downscaled environmental limits (please see Fig.S3-S7). To explicitly clarify this difference, we have edited the legend of the left-most panel in Fig. 1 and revised the legend of Fig. 4a.*

6. Results on Fig. 6b are not clear (discussion in lines 290-292). The figure is presented as the normalised version of 6a, but if that would be the case, then the figure would look exactly the same but with different values. How are the (normalized) values plotted in Fig. 6b calculated? Is it the average transgression (if any) among the five PBs considered? Please, clarify.

Response: *To address this comment, we have clarified the difference between panel a and panel b in Supplementary Information (please see Lines 501-504) and explained the calculation of environmental performance (x-axis) and human well-being (y-axis) (please see Lines 517-520). In response to the previous comment to focus on the key findings, we have also moved to this Figure to the Supplementary Information (instead of keeping it in the Main text)*

7. Discussion on Fig. 7 (lines 293-306) is not clear neither. Trade-off relationships are suggested between social well-being and all PBs except for climate change, but “similar” values are obtained by adding the percentage of regions in Quadrants I-III vs in Quadrants II-IV (e.g., 40% vs 50% for freshwater use, or 50% vs 43% for P cycle). In addition, percentages add up to 90% instead of 100% in Fig. 7 (please, check).

Response: Fig. 7 is now being moved to the Supplementary Information. In addition, we have also revised the discussion of Fig. S16 in the Supplementary Information (please see Lines 534-543). Further, to clarify the percentages of 100% and avoid confusions, we have added the percentages of points located on the axes (i.e., provinces with no trend) in Table. S15 and added Fig. S17 as a new figure.

8. Lines 184-188 could potentially contain a reference to the appropriate section of supplementary material. Similarly, section 4.4 on research limitations should include a reference to section 3 in the supplementary material, as well as briefly comment on the findings of this section. In general, all the parts in the supplementary material should be referred to somewhere in the main manuscript.

Response: We have included a reference to the part of Sensitivity analysis (Section 2.9 in Supplementary Information) in section 4.4 on research limitations and added some comment on these findings (please see Lines 635-639). In addition, we have referred other parts in the Supplementary Information in the main manuscript, including supplementary methods and results.

9. Conclusions are slightly vague in terms of significance and would benefit from including specific examples on the trends observed and guidelines provided. This way, they will demonstrate how the proposed framework can inform policymakers and help them develop more effective policies. This could also help clarifying the novelty of the manuscript.

Response: Thanks for your constructive comments! We have now largely revised the Conclusions Section. To address this comment, we have added the specific examples to explain how our proposed framework can help policymakers to develop effective policies (please see Lines 670-680). In addition, we have elaborated the novelty and contributions of our research in Lines 681-688.

10. While the explanation in lines 21-43 of the Supplementary Material is interesting, it still does not reflect on the rationale behind selecting only four (five) of the nine PBs quantified to date. Please, address this issue.

Response: To address this suggestion, we have now added the justification for selecting these four PBs in Lines 42-54 in Supplementary information. In addition, we have added a new section in Supplementary Information that explicitly stated the reasons why other PBs cannot be downscaled to the national scale in the Part of “1.2.5 Other boundaries” (please see Lines 186-207).

11. Results in Supplementary Fig. 2 are only described, but one would expect that their *implications* were also addressed. For instance, what is the impact of the high sensitivity of climate change and the P cycle in Supplementary Fig. 2a? Does this

change any of the conclusions in the main manuscript? This would not be a negative thing per se, but would simply deserve discussion. The same applies to results in Supplementary Fig. 2b.

Response: Thanks for your good suggestion! The sensitivity analysis in our study aims to address how sensitive results are to the downscaling method used. Our sensitivity analysis does not change any of conclusions in the main manuscript, but only serves as a proof of the robustness of the downscaling method used. We have added the implications of the results in Supplementary Fig. 21 in Lines 624-626 and 631-638.

Minor comment and typos:

12. Line 477, should read [...] is “in” contrast to [...].

Response: Many thanks! We have revised it in Line 514.

13. Avoid contract forms in formal/technical language (see “doesn’t” in lines 684 or “It’s” in line 685).

Response: We have revised the text in Lines 771 and 772, and checked other mistakes.

14. Data availability: Tables S14-S16 should also be referred to in the specific points of the manuscript where these data is relevant (e.g., Table S16 somewhere in lines 760-771).

Response: We have added the reference to the data sources in Lines 714, 755, and 854 in our manuscript.

15. Supplementary Figs. 3 and 4. Please indicate clearly the variables depicted in the Y-axes and their units.

Response: We have clarified the variables in the y-axes and their units in the captions of Fig. S12 and S13.

Suggestions:

16. Lines 178-181, clarify that two PBs are defined for biogeochemical flows, as otherwise the unfamiliar reader might be confused when reading “four planetary boundaries” and then finding five sectors in Fig. 3. The same applies to lines 636-638 in the Methods section (and other places in the manuscript).

Response: We have clarified this point in manuscript (please see Lines 241-243 and 710-712) and Supplementary Information (please see Lines 52-53).

17. In Fig. 3, I would suggest using consistent colours between social and environmental aspects: if grey is “bad” for social, let it be “bad” for the environment,

too (i.e., paint in green only the PBs which are not transgressed, and leave grey for those transgressed).

Response: Many thanks! We have revised the color of Fig. 3 and clarified the meanings of color in the figure caption.

18. In lines 331-339, briefly mention that the “V1-9 code” refers to the combination of different coordination degrees with social/environmental lag.

Response: We have added the explained the “V1-9 code” in Lines 323-325.

19. The discussion about the Balanced strategy (starting at line 513) is insightful and interesting. When talking about energy exporters regions, it might be worth mentioning that a different picture could emerge by measuring impacts through a consumption-based approach as done in previous works (e.g., Davis and Caldeira, PNAS. 2010, 107, 5687-5692; Pozo et al., JoCP 2020, 270, 121828).

Response: Thanks for this suggestion on the importance of comparison between different approaches to measure impacts. We have added this comparison in Lines 560-564 and cited the literature in Line 1021-1027.

20. Supplementary Table 13 could be more informative if presented as a matrix.

Response: Many thanks! We have modified Table. S5 into a matrix.

Reviewer #3 (Remarks to the Author):

This work has potential for a very interesting analysis of regional differences in environmental performance and social development and their driving factors in China. However, when reading the revised version, I don't see that the authors would have improved the manuscript and their analysis significantly enough for me to be able to recommend it as it is for publication in Nature Communications. My main points are the following:

1. Motivation of the study - The Introduction lacks a clear statement of the research challenge/question. It only lists objectives of the paper (i.e. what is being done). Because of this, it remains unclear for the reader what the purpose of the paper is and what new knowledge this paper aims to produce.

Response: Thanks for your great suggestion! We agree that it is important to clarify the research questions in the broader context of the literature, in addition to our outlined research objectives. To address this comment, we have substantially rewritten the Introduction section (please see Lines 47-153) that first lays out the importance and conceptual foundation of this work (i.e., sustainable development, SJOS, and strong sustainability concept). Second, we further pointed out the need of

developing an operationalizable framework to quantify sustainability patterns that accounts for both environmental limits and human wellbeing outcomes. Further, we elaborated the novel aspects of this research (as compared to previous work) that account for: (1) explicitly addressing spatial and temporal variations and across-scale dynamics in assessing sustainability relative to SJOS; (2) introducing an innovative coupling coordination degree analysis based on the sustainability assessment to conceptualize and account for interactions between environmental performance and human well-being goals; and (3) taking the SDGs as justifiable target thresholds for social indicators in assessing sustainability on the basis of SJOS.

2. Frameworks - Related to the unclear aim, it is also unclear why the authors have chosen the planetary boundaries framework for their study. Is it really the best framework if you are looking at how regional/provincial development and environmental impacts are connected, in particular if you are taking a territorial/production-based approach for estimating the environmental impacts?

Response: *Thanks for your great suggestion. To address this comment, we have re-stated the purpose of our research in the Introduction section, further explained the reasons for choosing the PB framework (please see Lines 61-78). The key strengths of the PB framework compared with alternative frameworks are its comprehensiveness and identification of critical thresholds/limits that it is not data-driven and that absolute benchmarks are provided (Nykvist et al., 2013). Based on these absolute benchmarks, we can further use the downscaling approach to delimit the environmental limits for our study region. We chose to adopt the PB framework so that we can place our results in a global context and compare with our findings with other regions or nations.*

However, in the revised manuscript, we did include discussion on the potential limitations of PB framework as compared to territorial/production-based approach (please see Lines 619-629). We also provided complementary analysis in the revision to compare the results using consumptive and territorial performance in Supplementary information (Section 2.2).

Nykvist, B., et al. National Environmental Performance on Planetary Boundaries. Swedish Environmental Protection Agency (2013).

3. The exercise of downscaling the planetary boundaries framework to national/regional level is often helpful when assessing a country's contribution to the global environmental pressures, including trade-related impacts. Why are the authors not including environmental indicators related to air and water pollution and resource use?

Response: *Great points. We fully agree that other indicators (e.g., those related to air and water pollution and resource use) can be critical for regional sustainability and*

further enrich our analysis. To address this suggestion, we have attempted to incorporate additional metrics for assessment related to environmental quality, in reference to key aspects of national policy concerns in section 2.3 in Supplementary Information (please see Lines 397-471). By considering these additional aspects that are locally relevant and context-specific, we are more likely to link regional and global sustainability realistically and effectively and inform actionable policies that are locally and regionally relevant. In addition, we have explained this point in Research Limitations Section (please see Lines 611-619).

4. Data – It is also unclear if the authors use “territorial/production-based” or “consumption-based” values for the environmental footprints. In many previous planetary boundaries downscaling studies, both production and consumption-based footprints are calculated and compared to give a fuller picture of a country’s impact. It’s also unclear when the authors are using values from O’Neill et al (2018) paper (see Fig 3 in the manuscript) or values from other sources.

Response: *Thanks for your comments! In our study, we used territorial-based footprints to measure environmental performance. We fully agree that the jointly framing of multiple sustainability perspectives (i.e., consideration of both production and consumption-based estimates) can more comprehensively assess and reflect the actual human-induced environmental pressures. To address this suggestion, we have further attempted to compare the consumptive with territorial performance at the national scale in Supplementary Information (please see Section 2.2), and our results showed overall consistent pattern and China underperforms in results estimated from territorial perspective (as compared to consumptive approach). In the revised manuscript, we did include discussion on the potential limitations of PB framework as compared to territorial/production-based approach (please see Lines 619-629). In addition, we have clarified data sources in the capture of Fig. 3 in Line 1130.*

5. Analysis and results – The authors use several methods, including statistical analysis, coupling coordination model and boosted regression three method based on machine learning. More explanation of why the authors have designed and implemented the study as it is, instead of merely telling what they have done, would have been welcome. It is also hard to understand how the authors have used these methods. The analysis of the produced results could have been done much more in depth.

Response: *Thanks for your valuable suggestions! In our revised manuscript, we have made edits throughout to better connect our analytical approach with our research objectives. In addition, to better illustrate our results according to the methods we used, we have reorganized the structure of the Results section. To help readers to better understand, we have added the purpose of the discussion on each section, explained the reasons for our choice of these methods, and briefly described how we used them at the beginning of each section. For example, 3.1 section in Lines 214-*

216, 3.1.1 section in Lines 236-238, 3.1.2 section in Lines 279-283, 3.2 section in Lines 308-312, 3.2.1 section in Lines 314-316, 3.2.2 section in Lines 365-374, 3.3 section in Lines 403-410.

6. Discussion - the Discussion section should discuss the results better, their meaning and insights from them. And, in particular, the results from the machine learning method and related discussion about the driver factors of the different development patterns could be further developed much more. The Discussion section also does not adequately situate the findings into the existing literature in development and sustainability research.

Response: *Thank you for your suggestions to further improve the Discussion section. In the revision, we have substantially revised the Results and Discussion by including additional discussion on drivers of coupling coordination in Results section (please see Lines 375-401) and interpretation of the driving factors for development patterns and coupling coordination results in Discussion Section (please see Lines 480-497). Our discussion focuses on identifying the main drivers and how these drivers affect coupling coordination degrees.*

7. Writing and style – the overall writing, structuring and organisation of the text should be improved to make it easier for the reader to follow the argumentation, findings and conclusions.

Response: *Thanks for your comments! To better illustrate our work and in consideration of similar comments from other reviewer, we have largely reorganized the structure of the manuscript and Supplementary Information to streamline the logic flow from research needs to major findings and interpretations. We also performed major revisions where only the most important findings and display items are retained in the Main Text.*

REVIEWER COMMENTS

Reviewer #2 (Remarks to the Author):

This report is for the third review round of the manuscript "Assessing coupling interactions in a safe and just operating space for regional sustainability". At this point, the authors have successfully addressed all my previous comments. Only some minor suggestions are left, based on inserts from the last review round:

1. Lines 101-113 mention "few studies" doing this or that, but no reference to these studies is provided. Please, provide them. In addition, either split the lumped references in line 101 to indicate the contribution of each study (similarly as done later in lines 127-131), or retain only a subset of them (seven joint references is too much!)
2. In lines 340-342, the authors point that 21/30 provinces are social development lag type. This is a surprising result considering that section 3.1 revealed that the social performance in China was better than the global average (while the opposite was true for the environmental performance).
3. Lines 560-564 suggest that employing a consumption-based approach (among others) could draw a different picture for the performance of some Chinese regions. How does this statement reconcile with lines 627-629, suggesting a consistent pattern between production and consumption-based assessments in China? Indeed, Figs. S4-S7 suggest disparate quantitative results between production and consumption-based perspectives, yet these are not sufficient to alter the category (i.e., safe/uncertain/risk) of most results.

Reviewer #3 (Remarks to the Author):

Dear Authors, thank you for the revised manuscript and adding some of the missing elements.

I regret to say that I still find the manuscript very confusing to read and not ready for publication. The authors have now added a research question in the Introduction. "How to quantify and achieve the SDGs without exceeding critical thresholds to inform actionable policies and development strategies." However, I do not see them properly answering this question in the Results and Discussion. The authors have now used lots of space to describe the very general, bigger context and background that is already commonly known in the sustainability science community. The potential novelty of their research, i.e., analysis of the different development patterns in different parts of China is still done in a very superficial way. I find Figure S14 useful - it compares the social and environmental performance of Chinese regions. But I still have a hard time understanding the meaning of the "coupling-coordination" calculations - these results are described in Fig. 8. What is the difference between Figures 8 and S14?

As I pointed out in the last review round, the manuscript would have been strengthened by adding a proper discussion about the reasons and driving factors of the different development patterns as well as situating the findings into the existing literature in development and sustainability research. The authors have now included results of drivers behind coupling coordination degree (i.e. the degree being a measure of the sustainability based on the interactions between environmental performance and human well-being indicators). According to their analysis, "grassland area and Normalized Difference Vegetation Index (NDVI) contribute to an increase in coupling coordination. In contrast, urbanization rate pulls in the opposite direction." I find it very difficult to understand the meaning of this and moreover, if the manuscript aims at understanding how and why different regions have developed differently environmentally and socially, an amount of grassland does not seem adequate to answer this. However, the authors do not provide any explanation to this or critically discuss their findings.

Dear Reviewers,

We much appreciate your constructive and insightful comments and suggestions. We have now carefully and substantively revised the manuscript based on these suggestions. Please see our detailed revisions in the manuscript where all changes were highlighted in red font. We also provided our point-by-point responses to the comments below.

REVIEWER COMMENTS

Reviewer #2 (Remarks to the Author):

This report is for the third review round of the manuscript “Assessing coupling interactions in a safe and just operating space for regional sustainability”. At this point, the authors have successfully addressed all my previous comments. Only some minor suggestions are left, based on inserts from the last review round:

***Response:** Thank you for your positive feedback related to our previous revision and your constructive comments. Please see our detailed response and revisions below.*

1. Lines 101-113 mention “few studies” doing this or that, but no reference to these studies is provided. Please, provide them. In addition, either split the lumped references in line 101 to indicate the contribution of each study (similarly as done later in lines 127-131), or retain only a subset of them (seven joint references is too much!)

***Response:** To address this comment, we have revised the Introduction section and added the missing references (please see Lines 92-102). In addition, we have removed several references and retained only a subset of them (please see Line 89-90).*

2. In lines 340-342, the authors point that 21/30 provinces are social development lag type. This is a surprising result considering that section 3.1 revealed that the social performance in China was better than the global average (while the opposite was true for the environmental performance).

***Response:** Thanks for your valuable comments! In calculating the coupling coordination degree (CCD), social development lag type is derived by comparing environmental performance with human well-being (i.e., overall social performance). In our revised manuscript, we classified CCD values into five classes in Fig. 6.a and distinguished the social/environmental development lag types (i.e., environmental lag, social lag, and socio-environmental synchronization types) by comparing environmental performance and human well-being results in Table. S16. To clarify this problem, we have added the coupling coordination evaluation framework (please see Fig. S2 in Supplementary Information) and revised the details of the classification criteria (please see Table. S5 in Supplementary Information).*

3. Lines 560-564 suggest that employing a consumption-based approach (among others) could draw a different picture for the performance of some Chinese regions. How does this statement reconcile with lines 627-629, suggesting a consistent pattern between production and consumption-based assessments in China? Indeed, Figs. S4-S7 suggest disparate quantitative results between production and consumption-based perspectives, yet these are not sufficient to alter the category (i.e., safe/uncertain/risk) of most results.

Response: To address this comment, we have removed this sentence and added specific statements about the differences between production and consumption-based assessment results in section 4.4 in manuscript (please see Lines 624-626).

Reviewer #3 (Remarks to the Author):

Dear Authors, thank you for the revised manuscript and adding some of the missing elements.

1. I regret to say that I still find the manuscript very confusing to read and not ready for publication. The authors have now added a research question in the Introduction. “How to quantify and achieve the SDGs without exceeding critical thresholds to inform actionable policies and development strategies.” However, I do not see them properly answering this question in the Results and Discussion. The authors have now used lots of space to describe the very general, bigger context and background that is already commonly known in the sustainability science community. The potential novelty of their research, i.e., analysis of the different development patterns in different parts of China is still done in a very superficial way.

Response: Thank you for your thoughtful comments and suggestions. In our substantially revised manuscript, we have now further clarified and highlighted the novelty of this research, which contributes to addressing three identified gaps in the literature: (1) transferrable downscaling; (2) spatial-temporal dynamics; and (3) SDG interactions. Conceptually, our study enriches the application and operationalizability of SJOS across a range of scales and social-ecological contexts by demonstrating its use in sustainable assessment that accounts for spatial-temporal heterogeneity and complex interactions between environmental performance and human wellbeing outcomes. Practically, our findings will help identify hotspots for target policy actions as well as assess progress towards achieving SDGs. Second, in our revised Introduction, per reviewer suggestion, we have also removed all the general, bigger context and background information commonly known in the sustainability science. In addition, we have added the results and discussion on driving factors and reasons for different development patterns (please see section 4.2 in manuscript and section 2.8 in SI). Further, we also modified our research question in the Introduction so that it better reflects the main findings of this work.

2. I find Figure S14 useful - it compares the social and environmental performance of Chinese regions. But I still have a hard time understanding the meaning of the “coupling-coordination” calculations - these results are described in Fig. 8. What is the difference between Figures 8 and S14?

Response: In our study, the results of environmental and social performance are the main input data for the calculation of coupling coordination degree (CCD). To better clarify CCD calculations, we have added the coupling coordination evaluation framework (please see Fig. S2) and details of calculation (please see Lines 319-324) in Supplementary Information.

Fig. S14 compares the social and environmental performance of Chinese regions and analyzes the relationships between them. Fig. 8 (i.e., Fig. 7.a in our revised manuscript) is mainly used to classify the regions into four different development patterns based on the coupling level (i.e., magnitude of CCD from Fig. 6.a) and development level (i.e., changing trends of CCD from Fig.

6.b) levels. Further, targeted strategies are recommended according to the characteristics and drivers of each pattern (please see Fig. 8 and Fig. S21). To clarify the differences these two figures, we have added the meaning and origin of x and y axes of Fig. 7.a in the caption (please see Lines 1159-1161) and main text (please see Lines 397-398).

3. As I pointed out in the last review round, the manuscript would have been strengthened by adding a proper discussion about the reasons and driving factors of the different development patterns as well as situating the findings into the existing literature in development and sustainability research. The authors have now included results of drivers behind coupling coordination degree (i.e. the degree being a measure of the sustainability based on the interactions between environmental performance and human well-being indicators). According to their analysis, “grassland area and Normalized Difference Vegetation Index (NDVI) contribute to an increase in coupling coordination. In contrast, urbanization rate pulls in the opposite direction.” I find it very difficult to understand the meaning of this and moreover, if the manuscript aims at understanding how and why different regions have developed differently environmentally and socially, an amount of grassland does not seem adequate to answer this. However, the authors do not provide any explanation to this or critically discuss their findings.

***Response:** Thanks for your valuable comments and thoughtful questions that allowed us to provide an in-depth discussion to different development patterns. Based on these suggestions and considerations, we have added the analysis of the drivers of changes in CCD for different development patterns and have substantially discussed to figure out what drives the differences of CCD in different development patterns (please see Fig. 8 in manuscript and Section 2.8 in SI). In addition, we have further proposed targeted strategies based on the characteristics of each pattern and their underlying drivers and causes and situated our findings into the literature in development for each pattern in Section 4.2 in manuscript (please see Lines 517-588).*

In addition, we have provided additional explanation of the results related to grassland area and NDVI (grounded upon the literature) in Section 4.1 in manuscript (please see Lines 457-489).